# Pathogenic POGZ mutation causes impaired cortical development and reversible autism-like phenotypes

Kensuke Matsumura et al.[#]

Pogo transposable element derived with ZNF domain (*POGZ*) has been identified as one of the most recurrently de novo mutated genes in patients with neurodevelopmental disorders (NDDs), including autism spectrum disorder (ASD), intellectual disability and White-Sutton syndrome; however, the neurobiological basis behind these disorders remains unknown. Here, we show that POGZ regulates neuronal development and that ASD-related de novo mutations impair neuronal development in the developing mouse brain and induced pluripotent cell lines from an ASD patient. We also develop the first mouse model heterozygous for a de novo *POGZ* mutation identified in a patient with ASD, and we identify ASD-like abnormalities in the mice. Importantly, social deficits can be treated by compensatory inhibition of elevated cell excitability in the mice. Our results provide insight into how de novo mutations on high-confidence ASD genes lead to impaired mature cortical network function, which underlies the cellular pathogenesis of NDDs, including ASD.

Neurodevelopmental disorders (NDDs), including autism spectrum disorder (ASD) and intellectual disability (ID), are characterized by early life onset with aberrant brain development, leading to social and cognitive abnormalities that span a wide range of functions and are highly heterogeneous among individuals[1,2]. Since the prevalence rate of NDDs has recently been increasing dramatically, NDDs are becoming a significant medical and social burden. Among NDDs, the prevalence rate of ASD is considerable, estimated to be ~1 in 40 children[3,4]. Despite the high heritability of ASD, recent studies have suggested that the disease is highly genetically heterogeneous, with rare genetic variants as well as common variants[5–7], and that the genetic cause is unidentified in ~90% of patients with the condition[8,9]. Accordingly, the molecular etiology of ASD remains largely unclear. There are no pharmacological medications to treat the core symptoms of ASD; mechanism-based drug development and therapeutic strategies are imperative.

Genetic and epidemiological studies have suggested that de novo mutations, spontaneous rare mutations that appear in an affected child but not in the unaffected parents, significantly contribute to ASD[10–18] and that ~3–10% of ASD risk is explained by de novo mutations in exons[6,16,19]. Importantly, a recent, comprehensive exome analysis has identified 65 high-confidence ASD genes[20]. Although, among these high-confidence ASD genes, recent studies have found that haploinsufficiency of ARID1B or CHD8 causes ASD-related abnormalities in mice[21–27], further direct assessments of the biological significance of ASD-associated de novo mutations are necessary to fully understand the contribution of de novo mutations to the core features of ASD.

We and other groups have found that Pogo transposable element derived with ZNF domain (POGZ) is one of the most recurrently mutated genes in patients with NDDs, particularly ASD and ID; the number of reported mutations continues to increase (see Fig. 1a and Supplementary Table 1; we classified patients into ASD, ID, and White–Sutton syndrome according to the original diagnosis in each report)[12,14–16,20,28–34]. POGZ mutations are also recurrently found in patients with White–Sutton syndrome, characterized by ID and specific facial features[30,31]. POGZ interacts with the SP1

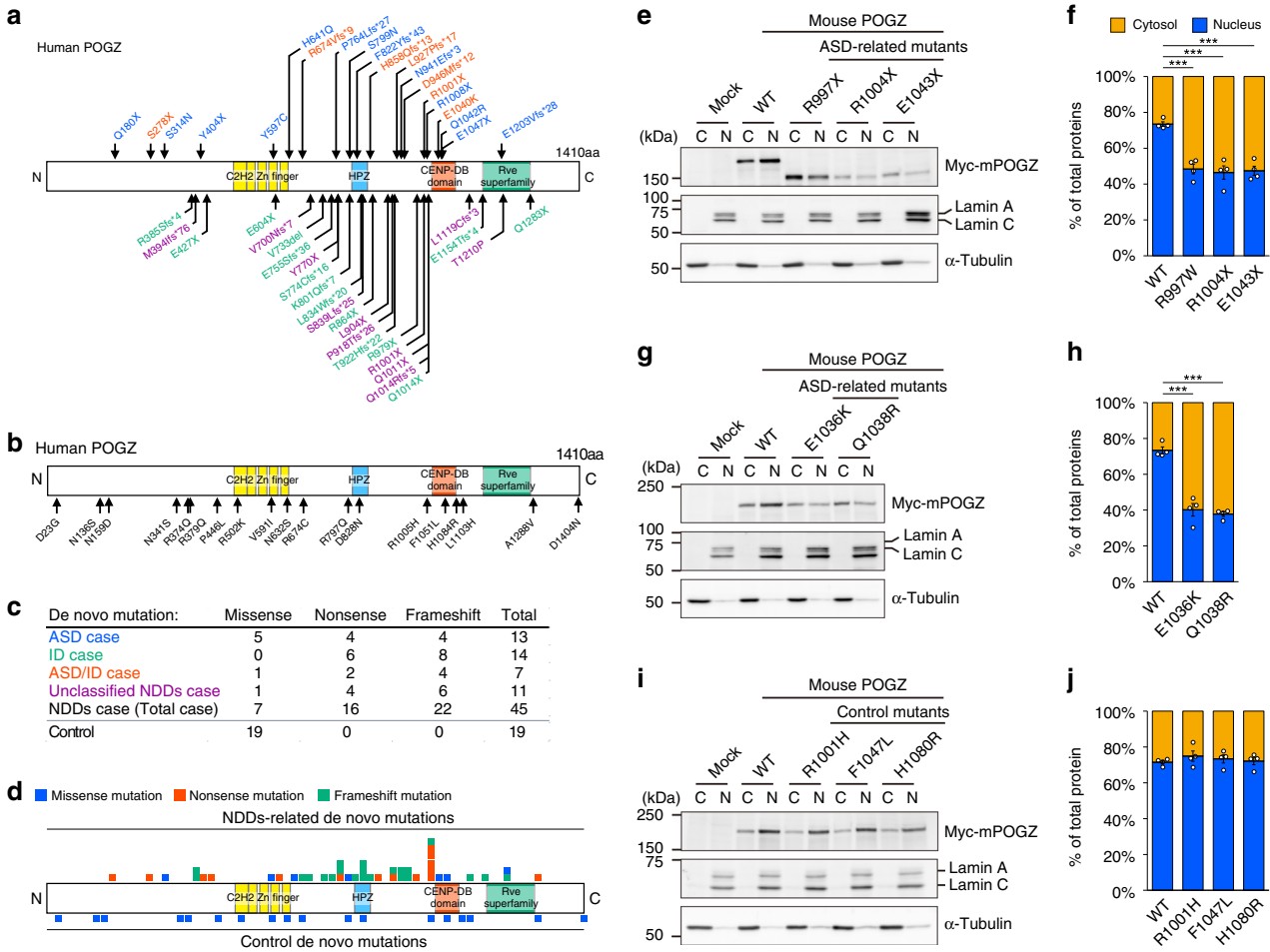

**Fig. 1 ASD-related de novo mutations in POGZ impair the nuclear localization of the POGZ protein. a** De novo mutations in POGZ identified in patients with NDDs (blue, ASD; green, ID; red, both ASD and ID; purple, unclassified NDDs). **b** De novo mutations in POGZ identified in unaffected controls. **c** The number of missense, nonsense, and frameshift mutations identified in patients with NDDs and unaffected controls. **d** Distribution of POGZ mutations identified in patients with NDDs and unaffected controls (blue, missense mutations; red, nonsense mutations; green, frameshift mutations). **e**, **g** ASD-related nonsense **e** or missense **g** mutations disrupted the nuclear localization of Myc-tagged overexpressed mouse (m) POGZ in Neuro2a cells. C cytosolic fraction; N nuclear fraction; WT wild-type. Note that Q1038R mutation in mouse POGZ corresponds to Q1042R mutation in human POGZ. **f**, **h**, **j** Quantification of POGZ in cytosolic and nuclear fractions (each $n = 4$). **i** Control mutations identified in unaffected healthy controls did not affect the nuclear localization of Myc-tagged overexpressed mouse POGZ in Neuro2a cells. C cytosolic fraction; N nuclear fraction; WT wild-type. Note that the amino acid numbers are based on the human protein (NP_055915.2) **a**, **b** and mouse protein (NP_766271.2) **e**, **g**, **i**. **f**, **h**, **j** One-way ANOVA with Bonferroni–Dunn post hoc tests, **f** $F_{3, 12} = 23.12$; **h** $F_{2, 9} = 67.45$; **j** $F_{3, 12} = 0.491$. ***$P < 0.001$. Data are presented as the mean ± s.e.m. The source data underlying figures **e**, **g** and **i** are provided as a Source Data file.

transcription factor, heterochromatin protein 1 (HP1), and chromodomain helicase DNA-binding protein 4 (CHD4)[35–37], which suggests that POGZ functions as a chromatin regulator; however, the role of POGZ in brain development and the biological significance of ASD-associated de novo POGZ mutations in the etiology of ASD are largely unknown.

In this study, we developed the first mouse model that carried a pathogenic de novo mutation of POGZ identified in an ASD patient. From the same patient, we established induced pluripotent stem cell (iPSC) lines with the same de novo POGZ mutated (Q1042R) as the model mouse. Comprehensively examining these human and mouse materials, we determined that the de novo mutation in POGZ impaired the cellular localization of the POGZ protein and hindered cortical neuronal development. We also determined that this de novo mutation in POGZ caused ASD-related behavioral abnormalities and that these abnormalities were pharmacologically treatable even in adulthood. Importantly, de novo POGZ mutations identified in unaffected controls had no damaging effect on POGZ function in neuronal development. Together, these observations provide the first in vivo evidence suggesting that ASD-associated de novo mutations in a high-confidence ASD gene are critical for a wide range of processes involved in ASD pathogenesis.

## Results

### De novo mutations in POGZ impair its nuclear localization.
POGZ has been identified as one of the most recurrently de novo-mutated genes in patients with NDDs (Fig. 1a, c, d, Supplementary Table 1; the amino acid numbers are based on the human protein). The vast majority of de novo POGZ mutations identified in patients with NDDs are nonsense and frameshift mutations and distributed between the C2H2 Zn finger and centromere protein-B-like DNA-binding (CENP-DB) domains and in the CENP-DB domain itself (Fig. 1a, c). In contrast, all de novo POGZ mutations identified in unaffected controls (control de novo mutations) are missense mutations (Fig. 1b, c, Supplementary Table 1). Interestingly, while control de novo missense mutations are uniformly distributed in POGZ, many NDDs-related nonsense and frameshift mutations are positioned just upstream of the CENP-DB domain, implying that impaired CENP-DB domain function could contribute to the risk of NDDs, including ASD (Fig. 1d). We examined the deleterious effect of sporadic-ASD-associated de novo missense mutations within the CENP-DB domain and nonsense mutations resulting in the elimination or truncation of the CENP-DB domain. Since the amino acid sequences of the human and mouse POGZ are very similar (93.9% identified in amino acid sequence) (Supplementary Fig. 1), we think that each mouse mutation is likely to correspond to the respective human mutation. Previous studies have suggested that POGZ is localized to the nucleus and functions as a chromatin regulator, we therefore assume that the ASD-related de novo mutations may alter the nuclear localization of POGZ. To examine this possibility, we firstly conducted immuno-cytochemical experiments using ASD-related missense mutants, E1036K (E1040K in human POGZ)- and Q1038R (Q1042R in human POGZ)-mutated POGZ, as well as E1043X (E1047X in human POGZ)-mutated POGZ, the longest nonsense-mutated POGZ, and found that these mutations partially impaired the nuclear localization of POGZ (Supplementary Fig. 2). We next performed cellular fractionation experiments and determined that, in contrast to the nuclear localization of overexpressed wild-type (WT)-mouse (m) POGZ, R997X (R1001X in human POGZ)-, R1004X (R1008X in human POGZ)-, and E1043X (E1047X in human POGZ)-mPOGZ mutants, which entirely or partially

lack the CENP-DB domain exhibited aberrant distribution in the cytoplasm (Fig. 1e, f; the amino acid numbers are based on the mouse protein). We also observed that the E1036K (E1040K in human POGZ)- and Q1038R (Q1042R in human POGZ)-mPOGZ mutants also exhibited aberrant distribution in the cytoplasm (Fig. 1g, h; the amino acid numbers are based on the mouse protein). Interestingly, in contrast to the ASD-related mutants, the de novo R1005H-, F1051L-, and H1084R-mPOGZ mutants, which harbor missense mutations within or adjacent to the CENP-DB domain identified in unaffected controls (Fig. 1b, d), showed similar protein expression patterns to WT-mPOGZ (Fig. 1i, j; the amino acid numbers are based on the mouse protein). Additionally, we performed cellular fractionation experiments using human SH-SY5Y cells and the human Q1042R-mutated POGZ and obtained essentially the same results as the results with the mouse Q1038R mutation in Fig. 1g, h (Supplementary Fig. 3). These results suggest that ASD-related de novo mutations but not control de novo mutations identified in unaffected controls impair the nuclear localization of POGZ in cells.

### POGZ regulates the development of mouse neural stem cells (NSCs).
To elucidate the function of POGZ in the brain, we first investigated the temporal, regional, and cell type-specific expression pattern of Pogz in the mouse brain. Temporally, expression of Pogz gradually increased during embryonic neurogenesis from embryonic day 14.5 (E14.5) to E18.5 and began to decrease after birth (Fig. 2a). At E16.5, Pogz was highly expressed in the cortical NSCs and intermediate progenitor cells (IPs) in the ventricular and subventricular zones (VZ/SVZ) (Fig. 2b, c). These expression patterns suggest that POGZ plays an important role in cortical neuronal development. To determine the role of POGZ in cortical neuronal development, we knocked down the expression of Pogz using four distinct commercial shRNAs (MISSION TRC shRNA library SP1, SIGMA-Aldrich) and a miR30-based shRNA (shRNA^miR30) targeting Pogz (Supplementary Fig. 4a, Supplementary Tables 2 and 3). Plasmids encoding each shRNA against Pogz and GFP were coelectroporated into the lateral ventricle of E14.5 mouse forebrains. The electroporated embryos were allowed to develop until E18.5 and histologically analyzed for migration of GFP⁺ cells in the developing somatosensory cortex. We determined that the migration of GFP⁺ cells was significantly inhibited by Pogz knockdown, which was roughly proportional to the knockdown efficiency of each construct (Supplementary Fig. 4b–i). The impaired migration was rescued by forced expression of WT-mPOGZ (Fig. 2d, e). Using antibodies against cortical layer markers, we then immunostained GFP⁺ cells whose migration was delayed by Pogz knockdown, and we determined that Pogz knockdown had little effect on the proportion of SATB2⁺ GFP⁺ (layer II/III), CTIP2⁺ GFP⁺ (layer V) or TBR1⁺ GFP⁺ (layer VI) neurons and that the GFP⁺ cells with delayed migration were mostly SATB2⁺ neurons, representing neurons with young upper cortical characteristics (Fig. 2f–k). Considering the high expression of Pogz in NSCs during cortical neuronal development (Fig. 2b, c), the delaying of migration by Pogz knockdown may be due to impaired neuronal differentiation. We analyzed the proportion of GFP⁺ NSCs, IPs, and neurons at E16.5 (2 days after in utero electroporation) and determined that Pogz knockdown increased the proportion of PAX6⁺ NSCs and decreased the proportion of TBR2⁺ IPs and SATB2⁺ young neurons without significantly affecting migration in the somatosensory cortex within 2 days (Fig. 3a–h). These data suggest that POGZ regulates cortical neuronal development by promoting neuronal differentiation.

### POGZ mutations impair POGZ function in neuronal development.
We investigated the effect of ASD-related de novo

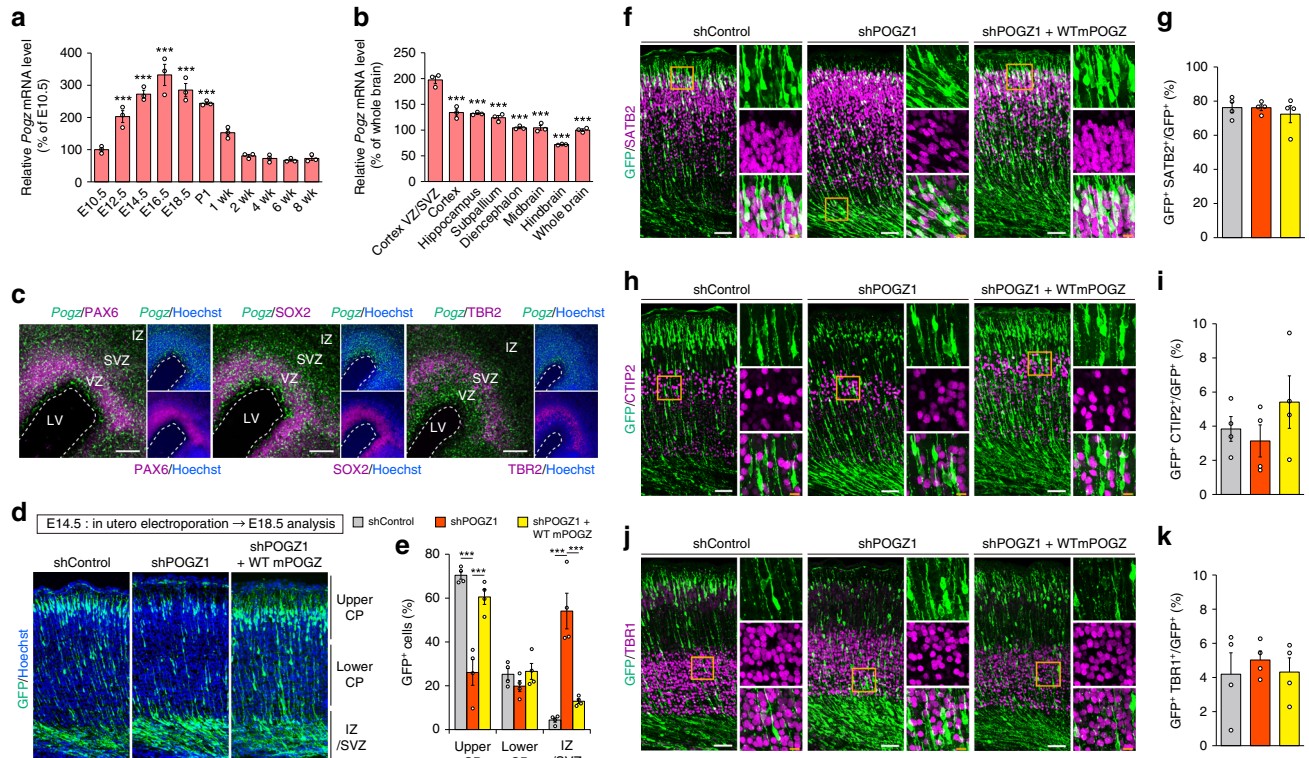

**Fig. 2 POGZ regulates the mouse cortical neuronal development. a** Temporal expression patterns of *Pogz* mRNA in the brain (each $n = 3$). E embryonic day; wk week-old. **b** Regional expression pattern of *Pogz* mRNA in the E16.5 brain (each $n = 3$). VZ ventricular zone; SVZ subventricular zone. **c** Fluorescence in situ hybridization of coronal sections of the E16.5 brain for *Pogz* and immunostaining with antibodies against PAX6, SOX2, and TBR2. Scale bars, 100 µm. LV lateral ventricle; VZ ventricular zone; SVZ subventricular zone; IZ intermediate zone. **d** Impairment of neuronal migration by shRNA-mediated knockdown of *Pogz* in E18.5 mouse cortices electroporated at E14.5. CP cortical plate; IZ intermediate zone; SVZ subventricular zone. Scale bars, 50 µm. **e** Quantification of GFP$^+$ cells in each layer (each $n = 4$). **f** GFP$^+$ neurons were co-labeled for SATB2 (a layer II/III marker). **g** Quantification of GFP$^+$ SATB2$^+$ neurons ($n = 4$). **h** GFP$^+$ neurons were co-labeled for CTIP2 (a layer V marker). **i** Quantification of GFP$^+$ CTIP2$^+$ neurons ($n = 4$). **j** GFP$^+$ neurons were co-labeled for TBR1 (a layer VI marker). **k** Quantification of GFP$^+$ TBR1$^+$ neurons ($n = 4$). Note that *Pogz* knockdown had little effect on the proportion of SATB2$^+$ GFP$^+$, CTIP2$^+$ GFP$^+$ or TBR1$^+$ GFP$^+$ neurons and that the GFP$^+$ cells with delayed migration were mostly SATB2$^+$ neurons. **h** Same slice as in **f**. **f**, **h**, **j** Right panels, magnifications of the areas outlined with orange boxes. White scale bars, 50 µm; orange scale bars, 10 µm. **a**, **b**, **g**, **i**, **k** One-way ANOVA with Bonferroni–Dunn post hoc tests; **a** $F_{10, 22} = 48.52$; **b** $F_{7, 16} = 73.23$; **g** $F_{2, 9} = 0.393$; **i** $F_{2, 9} = 1.079$; **k** $F_{2, 9} = 0.239$. **e** Two-way repeated-measures ANOVA with Bonferroni–Dunn post hoc tests, $F_{4, 27} = 39.28$. ***$P < 0.001$. Data are presented as the mean ± s.e.m.

mutations on embryonic cortical neuronal development using in utero electroporation gene delivery in E14.5 embryos. We determined that forced expression of the ASD-related mPOGZ mutants failed to rescue the *Pogz*-knockdown-mediated migration defect at E18.5 (Fig. 4a, b). In contrast to the ASD-related mPOGZ mutants, control mPOGZ mutants, including the R1005H-, F1051L- and H1084R-POGZ mutants, rescued the *Pogz*-knockdown-mediated migration defect to virtually the same level as WT-mPOGZ expression at E18.5 (Fig. 4c, d). Thus, the ASD-related de novo mutations, but not the control de novo mutations identified in unaffected controls, disrupt the function of POGZ in embryonic cortical neuronal development, implying the pathogenicity of the ASD-related de novo mutations in cortical neuronal development. We then performed overexpression experiments using ASD-related de novo mutated POGZ using WT embryos. We determined that the expression of R1004X-, E1036K-, Q1038R- and E1043X-mutated POGZ impaired the migration of GFP$^+$ cells in WT neurons (Supplementary Fig. 5), suggesting that the de novo mutations show a dominant-negative effect upon cell migration. Considering that the de novo-mutated POGZ showed reduced nuclear localization (Fig. 1), abnormally localized de novo-mutated POGZ in the cytoplasm might inhibit the function of endogenous POGZ (e.g., abnormally localized de novo-mutated POGZ may titrate the interaction partner of POGZ

in the cytoplasm). Alternatively or in addition, WT and de novo-mutated POGZ might compete each other in the nucleus.

**Neuronal differentiation is impaired in NSCs derived from a patient with sporadic ASD carrying a de novo Q1042R mutation of POGZ.** We investigated the effect of the ASD-related de novo mutation on NSCs derived from a patient with sporadic ASD. We previously recruited Japanese sporadic autism trios and identified an ASD patient carrying the Q1042R mutation of POGZ[15]. We established iPSC lines using immortalized B cells obtained from that patient and an unaffected healthy control (Supplementary Fig. 6) and differentiated each iPSC lines into NSCs. ASD-patient-derived and control NSCs were seeded into neuronal differentiation medium on day 0 of neuronal differentiation. Then the number of MAP2$^+$ neurons was analyzed at day 2, when they were in an early stage of neuronal differentiation. The proportion of MAP2$^+$ neurons was significantly lower in the patient-derived NSCs than in the control NSCs, suggesting that neuronal differentiation is impaired in the patient-derived NSCs (Fig. 4e, f). We then analyzed the self-renewal activity of the patient-derived NSCs. We determined that the neurospheres from patient-derived NSCs were larger than those from control NSCs and that the patient-derived NSCs exhibited higher

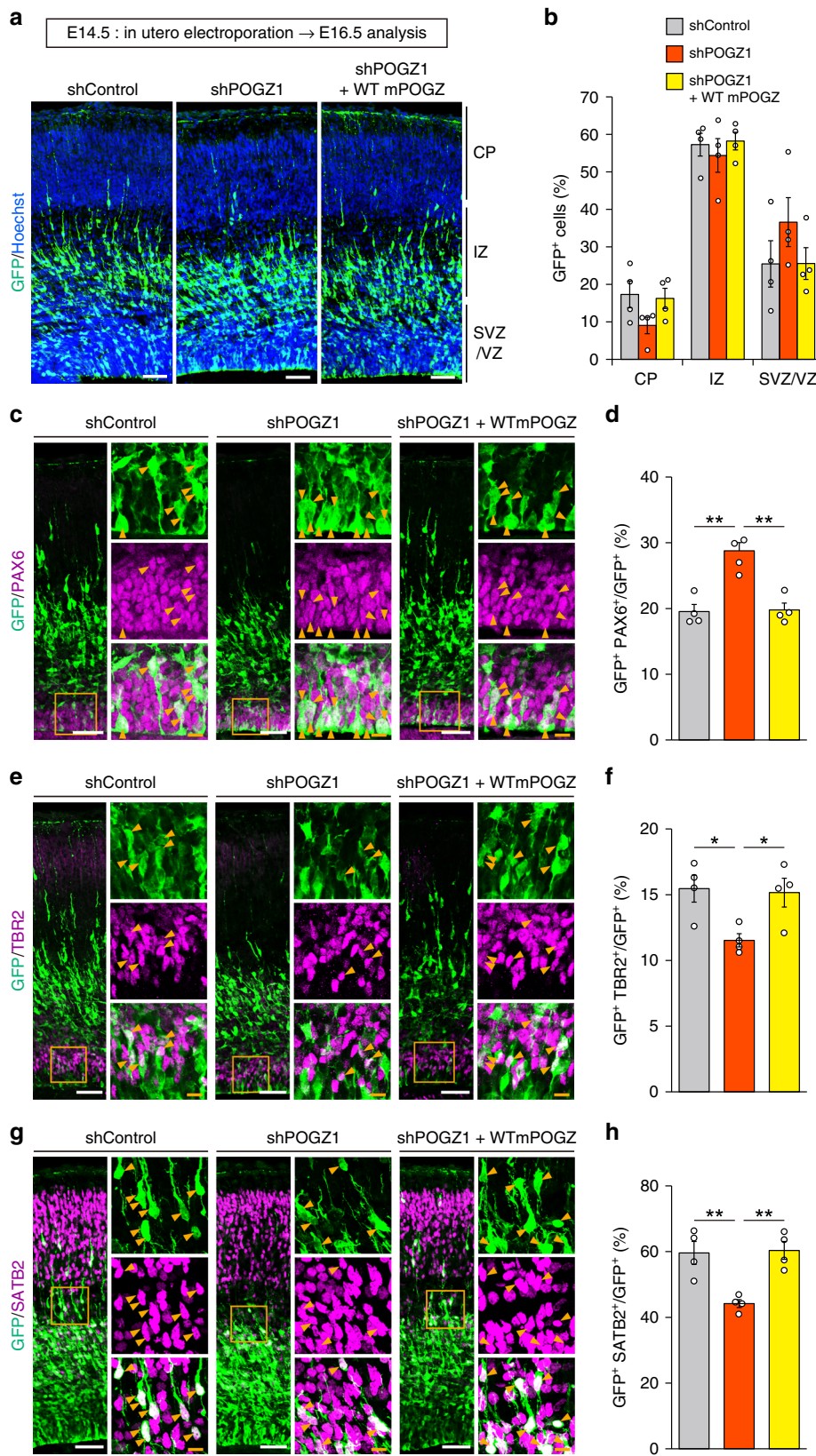

bromodeoxyuridine (BrdU) incorporation than the control NSCs (Supplementary Fig. 7 and Fig. 4g). Furthermore, we performed an in vitro migration assay in which newborn neurons migrate outwards radially from neurospheres after adhesion, and we determined that the radial migration of young neurons was significantly attenuated in the patient-derived NSCs (Fig. 4h, i). Taken together, these results show that, consistent with the impaired cortical neurogenesis induced by *Pogz* knockdown in mice (Fig. 4a, b), the neuronal differentiation is impaired in patient-derived NSCs.

**Fig. 3 POGZ regulates the neuronal differentiation of mouse cortical neural stem cells. a** Slight, non-significant migration defects caused by shRNA-mediated knockdown of *Pogz* in E16.5 mouse cortices electroporated at E14.5. CP cortical plate; IZ intermediate zone; SVZ subventricular zone; VZ ventricular zone; E embryonic day. Scale bars, 50 μm. **b** Quantification of GFP+ cells in each layer (each *n* = 4). CP cortical plate; IZ intermediate zone; SVZ subventricular zone; VZ ventricular zone. **c** Increased number of PAX6+ NSCs caused by *Pogz* knockdown in E16.5 mouse cortices electroporated at E14.5. **d** Quantification of PAX6+ cells (each *n* = 4). **e** Decreased number of TBR2+ differentiated IPs caused by *Pogz* knockdown in E16.5 mouse cortices electroporated at E14.5. **f** Quantification of TBR2+ cells (each *n* = 4). **g** Decreased number of SATB2+ differentiated neurons caused by *Pogz* knockdown in E16.5 mouse cortices electroporated at E14.5. **h** Quantification of SATB2+ cells (each *n* = 4). **c**, **e**, **g** Right panels, magnifications of the areas outlined with orange boxes. Arrowheads indicate co-labeled cells. White scale bars, 50 μm; orange scale bars, 10 μm. **b** Two-way repeated-measures ANOVA with Bonferroni–Dunn post hoc tests, $F_{4, 27} = 1.861$. **d**, **f**, **h** One-way ANOVA with Bonferroni–Dunn post hoc tests; **d** $F_{2, 9} = 18.37$; **f** $F_{2, 9} = 5.710$; **h** $F_{2, 9} = 11.91$. *$P < 0.05$, **$P < 0.01$. Data are presented as the mean ± s.e.m.

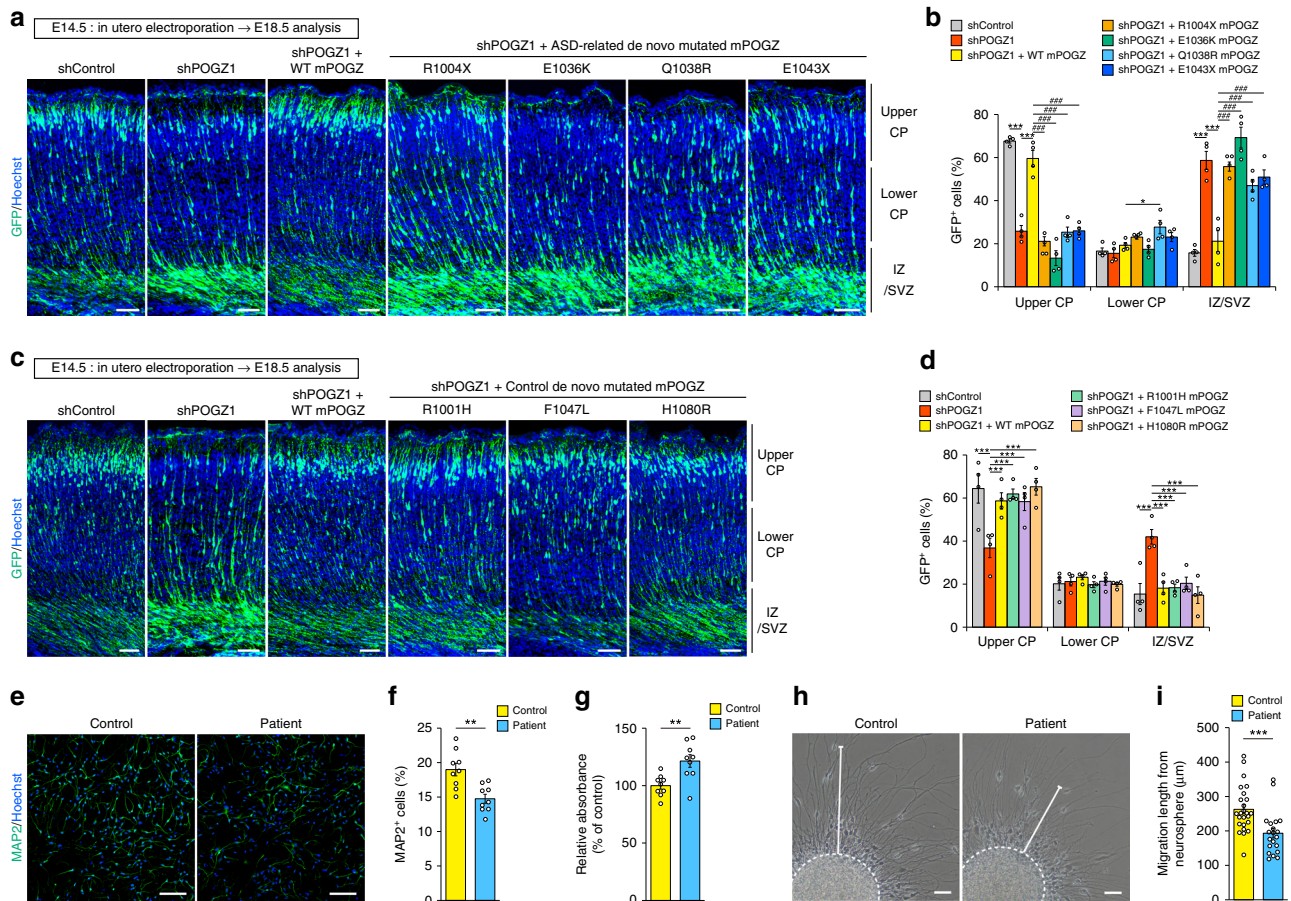

**Fig. 4 ASD-related POGZ mutations impair neuronal development. a** Forced expression of the ASD-related POGZ mutants failed to rescue the *Pogz*-knockdown-mediated migration defect in the developing mouse cortex. E embryonic day. Scale bars, 50 μm. **b**, **d** Quantification of GFP+ cells in each layer (each *n*=4). CP cortical plate; IZ intermediate zone; SVZ subventricular zone. **c** Forced expression of the control POGZ mutants rescued the *Pogz*-knockdown-mediated migration defect. Scale bars, 50 μm. **e** Low number of MAP2+ differentiated neurons in the patient-derived NSCs harboring the Q1042R POGZ mutation. Scale bars, 100 μm. **f** Quantification of MAP2+ neurons (each *n* = 9). **g** Increased BrdU incorporation in the patient-derived NSCs (each *n* = 9). **h** Impaired radial migration of the patient-derived neurons. Scale bars, 50 μm. **i** Quantification of the migration distance (control, *n* = 25; patient, *n* = 20). **b**, **d** Two-way repeated-measures ANOVA with Bonferroni–Dunn post hoc tests; **b** $F_{12, 63} = 54.55$; **d** $F_{10, 54} = 9.426$. **f**, **g**, **i** Student's *t*-test. **$P < 0.01$, ***$P < 0.001$, ###$P < 0.001$ (vs. shPOGZ1 + WT mPOGZ in **b**). Data are presented as the mean ± s.e.m.

## Generation of $POGZ^{WT/Q1038R}$ mice using CRISPR-Cas9 gene editing.

To reveal the functional significance of the ASD-related Q1042R mutation in brain development and behavioral characteristics, we generated $POGZ^{WT/Q1038R}$ mice heterozygous for the Q1038R mutation, corresponding to the human Q1042R mutation, using CRISPR-Cas9 gene editing (Supplementary Fig. 8a, b). Heterozygous $POGZ^{WT/Q1038R}$ mice were born in the ratio predicted by Mendelian genetics (Supplementary Fig. 8c), and they exhibited reduced body size and brain in adulthood compared to WT mice (Supplementary Fig. 8d–h). We next

measured the thicknesses of cortical layers in the somatosensory cortex and found that the thickness of layers II–IV and V in $POGZ^{WT/Q1038R}$ mice were slightly decreased and increased, respectively (Supplementary Fig. 8i–n). Although $POGZ^{WT/Q1038R}$ mice exhibited decreased brain size, we did not find any drastic histological abnormalities, such as heterotopias, in the cortex of $POGZ^{WT/Q1038R}$ mice. We histologically examined patient-related non-neurological abnormalities in adult $POGZ^{WT/Q1038R}$ mice and found that $POGZ^{WT/Q1038R}$ mice did not exhibit any significant changes in peripheral organs, including

eye, cochlea, trachea, stomach, duodenum, ileum, cecum and colon, compared to WT mice (Supplementary Fig. 9). Additionally, we did not find diaphragmatic hernia in adult $POGZ^{WT/Q1038R}$ mice. Furthermore, we performed micro-CT scanning of adult $POGZ^{WT/Q1038R}$ mice. We did not find any significant abnormalities in the skull of $POGZ^{WT/Q1038R}$ mice (Supplementary Fig. 10).

In addition, no homozygous point mutant ($POGZ^{Q1038R/Q1038R}$) offspring were produced by mating male and female $POGZ^{WT/Q1038R}$ mice (0 of 186 pups; Supplementary Fig. 11a). We performed micro-CT scanning of mouse embryos and found that $POGZ^{Q1038R/Q1038R}$ mouse embryos (E15.5) showed a ventricular septal defect, which likely results in embryonic lethality ($n = 4$) (Supplementary Fig. 11b).

**Embryonic cortical neuronal development is impaired in $POGZ^{WT/Q1038R}$ mice.** We examined embryonic cortical neuronal development and determined that the density of the $SATB2^+$ neurons (layer II/III) was decreased in the upper layer and increased in the lower layer in the developing somatosensory cortex (Supplementary Fig. 12a, b), indicating the abnormal distribution of $SATB2^+$ cortical excitatory neurons in $POGZ^{WT/Q1038R}$ mice at E18.5. To confirm the impairment of cortical neuronal development in $POGZ^{WT/Q1038R}$ mice, we labeled the new-born neurons with BrdU at E14.5. We determined that $POGZ^{WT/Q1038R}$ mice exhibited a decreased number of $SATB2^+$ $BrdU^+$ cells in the upper layer and an increased number of $SATB2^+$ $BrdU^+$ cells in the lower layer in the developing cortex (Supplementary Fig. 12c, d). These results suggest that $POGZ^{WT/Q1038R}$ mice exhibit impaired embryonic cortical neuronal development, which is consistent with the impaired neuronal development in the NSCs derived from the ASD patient carrying the de novo Q1042R mutation in POGZ (Fig. 4e–i). We also histologically examined the distribution of $CUX1^+$ cortical neurons (layer II/III) and determined that $CUX1^+$ excitatory neurons were still abnormally distributed in the adult $POGZ^{WT/Q1038R}$ mice (Supplementary Fig. 12e, f). There were no significant changes in the average density of the $CUX1^+$ neurons (WT, $1126 \pm 36.86$ cells per mm²; $POGZ^{WT/Q1038R}$, $1105 \pm 52.64$ cells per mm²). In contrast to excitatory neurons, the average density and distribution of $GABA^+$ interneurons were indistinguishable between WT and $POGZ^{WT/Q1038R}$ mice (Supplementary Fig. 12g, h; the average density of $GABA^+$ interneurons, WT, $121.6 \pm 3.710$ cells per mm²; $POGZ^{WT/Q1038R}$, $120.9 \pm 2.527$ cells per mm²).

**Transcriptional networks underlying neuronal development is altered in NSCs derived from both the ASD patient carrying the Q1042R mutation of POGZ and $POGZ^{WT/Q1038R}$ Mice.** Given that POGZ interacts with HP1 and CHD4[35] and is suggested to bind DNA, POGZ may modulate the neuronal differentiation of NSCs through regulation of gene expression. To examine this possibility, we performed RNA-sequencing on NSCs derived from the ASD patient carrying the de novo Q1042R mutation of POGZ and E16.5 embryonic cortex of $POGZ^{WT/Q1038R}$ mice (significant results are shown in Supplementary Tables 4 and 5). We then analyzed gene ontology (GO) annotation of the differentially expressed genes between the unaffected healthy control and patient, and WT and $POGZ^{WT/Q1038R}$ mice and found that the differentially expressed genes in human and mice were commonly enriched for GO annotations involving cellular and organismal development, particularly neuronal development (Supplementary Fig. 13a). In particular, 78 out of 913 and 251 genes annotated to neurogenesis (GO: 0022008) in human and mouse, respectively, showed commonly differential expression between human and mouse (Supplementary Tables 4 and 5). Considering that POGZ represses gene transcription in

hematopoietic cells[37,38], we focused on the upregulated genes in NSCs derived from the patient and $POGZ^{WT/Q1038R}$ mice. We found that, among these differentially expressed genes involving neuronal development, a Notch ligand, Jagged canonical Notch ligand 2 (*JAG2*), was expressed approximately two-fold higher in NSCs derived from both the patient (fold change = 1.970) and $POGZ^{WT/Q1038R}$ mice (fold change = 2.175) compared to each corresponding control NSCs (Supplementary Tables 4 and 5). To investigate whether POGZ binds to the *Jag2* promoter in NSCs, we performed chromatin immunoprecipitation (ChIP) assays using cortical NSCs derived from E16.5 WT mice. We found that chromosome containing the *Jag2* promoter was enriched by anti-POGZ antibodies, suggesting that POGZ binds to the *Jag2* promoter (Supplementary Fig. 13b, c). Together with the fact that Notch signaling negatively regulate neuronal differentiation of NSCs[39,40], these results suggest that POGZ may facilitate neuronal development by inhibiting gene expression, including *JAG2*.

**$POGZ^{WT/Q1038R}$ mice show ASD-related behavioral abnormalities.** We performed a series of behavioral tests, including open-field, home-cage activity, light/dark transition, Y-maze, fear conditioning, novel object recognition and prepulse inhibition (PPI) tests, in adult $POGZ^{WT/Q1038R}$ mice. We observed that $POGZ^{WT/Q1038R}$ mice showed slightly increased home-cage activity in the light phase and impaired cognitive function in the novel object recognition and fear conditioning tests (Supplementary Fig. 14a–g). In the open-field test, although the locomotion of WT and $POGZ^{WT/Q1038R}$ mice was indistinguishable in quantity, $POGZ^{WT/Q1038R}$ mice spent more time in the center zone than their WT littermates did (Fig. 5a, b). In the other tests, including the light/dark transition, Y-maze and PPI tests, $POGZ^{WT/Q1038R}$ mice showed behavioral features similar to those of their WT littermates (Supplementary Fig. 14h–k; latency for entering into light chamber in the light/dark transition test, WT, $119.4 \pm 69.1$ s, $POGZ^{WT/Q1038R}$, $47.4 \pm 13.7$ s, $P > 0.05$, Student's *t*-test). We then investigated social and repetitive behaviors in adult $POGZ^{WT/Q1038R}$ mice. In the reciprocal social interaction test, $POGZ^{WT/Q1038R}$ mice spent less time in sniffing novel mice than their WT littermates did, suggesting impaired social interaction (Fig. 5c). In the self-grooming test, $POGZ^{WT/Q1038R}$ mice spent more time in repetitive self-grooming than their WT littermates did, suggesting aberrant repetitive behavior (Fig. 5d). These data collectively suggest that adult $POGZ^{WT/Q1038R}$ mice show behavioral abnormalities associated with NDDs, including ASD. We next investigated whether the behavioral abnormalities of $POGZ^{WT/Q1038R}$ mice are present early in development; we determined that juvenile $POGZ^{WT/Q1038R}$ mice spent less time than their WT littermates in active interactions, such as sniffing, allogrooming, chasing, and playing, suggesting impaired social interactions in juvenile $POGZ^{WT/Q1038R}$ mice (Fig. 5e). Finally, we assessed the communication ability of $POGZ^{WT/Q1038R}$ pups at postnatal day 4 (P4) by measuring isolation-induced ultrasonic vocalization (USV) responses emitted by mouse pups separated from their mothers. After separation from their mothers, $POGZ^{WT/Q1038R}$ pups emitted more and longer USV calls than their WT littermates did (Fig. 5f, g). $POGZ^{WT/Q1038R}$ pups exhibited an altered call pattern, such as increased proportions of two-syllable and flat calls and decreased composite calls and frequency steps, suggesting a disturbance in communications from $POGZ^{WT/Q1038R}$ pups to their mothers (Fig. 5h). Thus, $POGZ^{WT/Q1038R}$ mice exhibit social deficits even in early development, which may effectively reflect the behavioral characteristics of NDDs, including ASD.

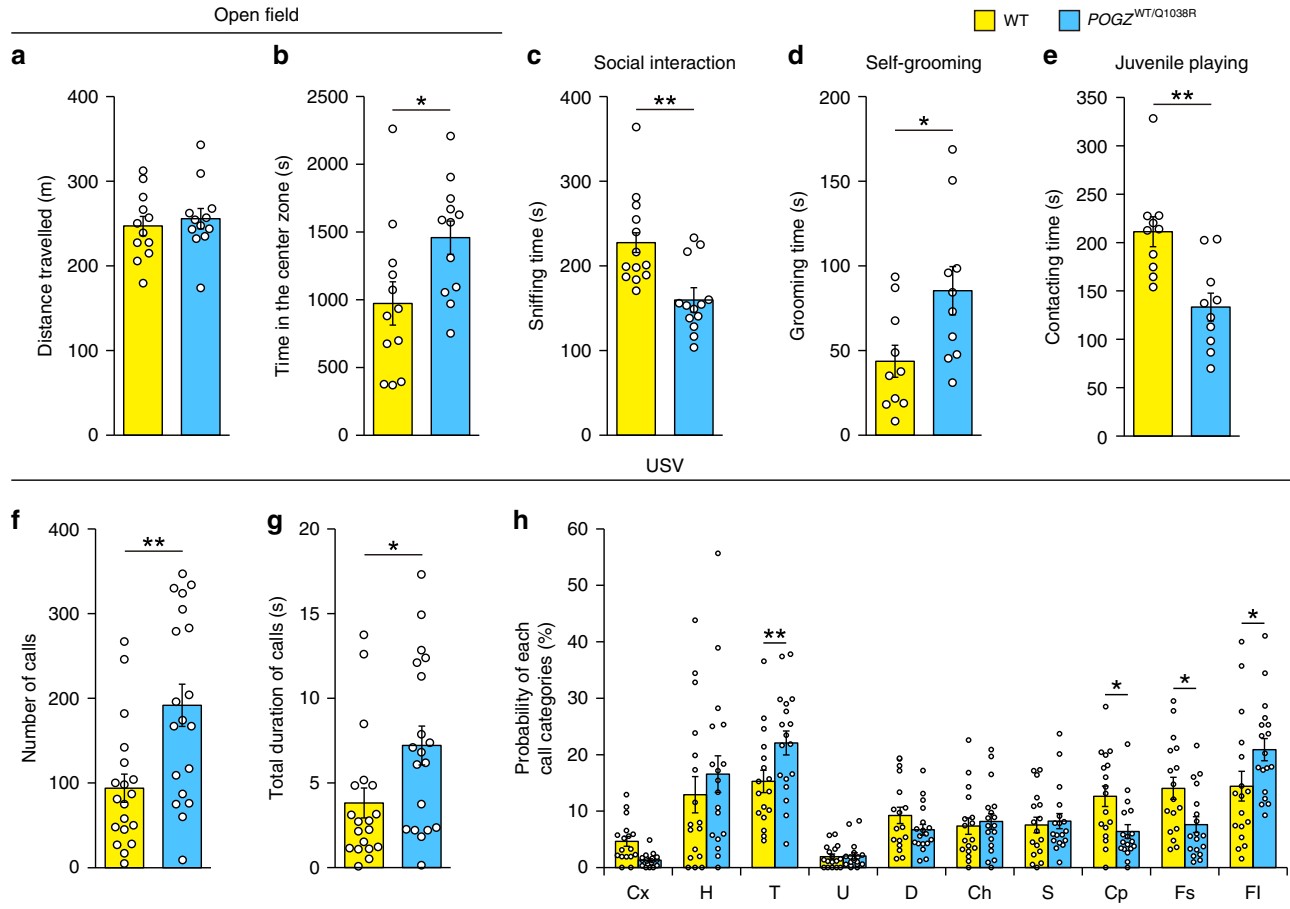

**Fig. 5 NDDs-related behavioral abnormalities in *POGZ*^WT/Q1038R mice. a** Distance traveled in the open-field test (each $n = 12$). **b** Time spent in the center zone in the open-field test (each $n = 12$). **c** Time spent sniffing in the reciprocal social interaction test (each $n = 13$). **d** Time spent grooming in the self-grooming test (each $n = 10$). **e** Time spent contacting in the juvenile playing test (each $n = 10$). **f** Numbers of ultrasonic calls made by WT and *POGZ*^WT/Q1038R mice at postnatal day 4 (each $n = 19$). USV ultrasonic vocalization. **g** Total duration of ultrasonic calls (each $n = 19$). **h** Altered ultrasonic call patterns in *POGZ*^WT/Q1038R mice (WT, $n = 17$; *POGZ*^WT/Q1038R, $n = 18$) (Cx complex; H harmonics; T two-syllable; U upward; D downward; Ch chevron; S shorts; Cp composite; Fs frequency steps; Fl flat). WT wild-type. **a–g** One-way ANOVA; **a** $F_{1, 22} = 0.277$; **b** $F_{1, 22} = 5.771$; **c** $F_{1, 24} = 13.02$; **d** $F_{1, 18} = 5.914$; **e** $F_{1, 18} = 13.62$; **f** $F_{1, 36} = 10.64$; **g** $F_{1, 36} = 5.465$. **h** Two-way repeated-measures ANOVA with Bonferroni–Dunn post hoc tests, $F_{9, 330} = 3.376$. $*P < 0.05$, $**P < 0.01$. Data are presented as the mean ± s.e.m.

**The activity of excitatory cortical neurons is increased in *POGZ*^WT/Q1038R mice.** Previous studies have suggested that an altered cellular balance of excitation and inhibition (E/I balance) within neural circuitry may cause the social and cognitive deficits that characterize ASD[41,42]. We next focused on whole-brain neuronal activation and suppression patterns during the social interaction test in *POGZ*^WT/Q1038R mice crossed with Arc-dVenus reporter mice that expressed the destabilized form of the fluorescent protein Venus (dVenus) driven by the promoter of the immediate early gene Arc[43]. Using FAST (block-FAce Serial microscopy Tomography), a high-speed serial-sectioning imaging system that we previously developed[44,45], we examined dVenus expression in the whole brains of 10-week-old WT and *POGZ*^WT/Q1038R mice in an unbiased manner; we determined that the activation of excitatory neurons in the anterior cingulate cortex (ACC), which is suggested to be involved in ASD[46], was higher in *POGZ*^WT/Q1038R mice than in WT mice after social interaction (Supplementary Fig. 15a–d). Principal component (PC) analysis of the normalized numbers of dVenus+ cells in WT and *POGZ*^WT/Q1038R mice revealed that WT and *POGZ*^WT/Q1038R mice were separated by PC 2 and that the ACC was a major contributor to PC2 (Supplementary Fig. 15d). We also observed that *POGZ*^WT/Q1038R mice, in comparison with WT mice,

had an increased density of dendritic spines in the pyramidal neurons in layer II/III of the ACC (Supplementary Fig. 15e, f). We then evaluated excitatory neurotransmission in 10-week-old *POGZ*^WT/Q1038R mice using whole-cell patch-clamp recordings from pyramidal neurons in layer II/III of the ACC. Whereas there were no changes in amplitude of miniature excitatory post-synaptic currents (mEPSCs) between WT and *POGZ*^WT/Q1038R neurons, the frequency of mEPSCs was drastically increased in *POGZ*^WT/Q1038R neurons compared to WT neurons (Supplementary Fig. 15g–k). Despite the decrease in the number of the excitatory pyramidal neurons in the upper cortical layer (Supplementary Fig. 12e, f), these results suggest that the excitatory neurons in the cerebral cortex is hyperactivated during the social interaction task in *POGZ*^WT/Q1038R mice.

**Treatment with an anti-epileptic agent, perampanel, improves the social deficits in *POGZ*^WT/Q1038R mice.** Given that *POGZ*^WT/Q1038R mice showed the elevated activation of excitatory neurons after social interaction and abnormally activated excitatory synaptic transmission, we investigated whether pharmacological inhibition of AMPA-mediated synaptic transmission could rescue the impaired social interaction typical of *POGZ*^WT/Q1038R mice. According to the previous studies, we

determined the minimum doses of NBQX and perampanel for antiepileptic activity. With these doses, we found that 10 mg/kg of NBQX did not affect the locomotor activity and that 3 mg/kg of perampanel tended to slightly decrease the locomotor activity in the open field, which is not statistically significant (Supplementary Fig. 16a, c). We intraperitoneally administered 10 mg/kg of NBQX, a competitive AMPA receptor antagonist, to $POGZ^{WT/Q1038R}$ mice 30 min prior to the reciprocal social interaction test and determined that NBQX treatment effectively rescued the time spent that $POGZ^{WT/Q1038R}$ mice spent sniffing intruder mice without affecting sniffing time in WT mice (Supplementary Fig. 16b). We also administered perampanel, a negative allosteric modulator of the AMPA receptor approved by the European Medicines Agency (EMA), the US Food and Drug Administration (FDA), and Japanese Pharmaceuticals and Medical Devices Agency (PMDA) for epilepsy treatment. Interestingly, oral administration of 3 mg/kg of perampanel successfully rescued sniffing time in $POGZ^{WT/Q1038R}$ mice (Supplementary Fig. 16d). These data suggest that the impaired social interaction observed in $POGZ^{WT/Q1038R}$ mice is likely to be caused by hyperactivation of excitatory synaptic transmission.

## Discussion

Analyzing functional mutations in individual putative causative genes for ASD is important for gaining mechanistic and pharmacological insights into ASD. Recent genetic and epidemiological studies suggest that the compromising of POGZ function by de novo mutations is likely to be involved in ASD; however, the contribution of de novo POGZ mutations to ASD onset remains largely unclear. In silico prediction shows that 17 out of 19 missense mutations identified in unaffected controls as well as 6 out of 7 NDD-related missense mutations are suggested to be damaging by at least one out of four predictive tools, namely, PROVEAN, SIFT, PolyPhen2, MutationTaster, CADD score, and The American College of Medical Genetics and Genomics (ACMG) classification (Table 1); in silico prediction of the pathogenicity of missense mutations is thus challenging and biologically assessing the pathogenicity of de novo mutations is important for understanding the etiology of ASD. Here, we assessed the pathogenicity of de novo POGZ missense mutations in vitro and in vivo and showed that the ASD-related de novo mutants, but not the control de novo mutants identified in unaffected controls, disrupt the nuclear localization of POGZ and embryonic cortical development. We also developed a new mouse model carrying a de novo POGZ mutation identified in an ASD patient. In addition to the model mouse, we established iPSC lines from the ASD patient with de novo-mutated POGZ. By comprehensively examining these human and mouse materials, we provide the first in vivo evidence suggesting that ASD-associated de novo mutations in a high-confidence ASD gene cause a wide range of aspects of the ASD phenotype.

In this study, using knockdown approaches, we provided the first in vivo evidence suggesting that POGZ regulates cortical excitatory neuron development by promoting neuronal differentiation (Figs. 2 and 3). Importantly, we determined that the neuronal developmental gene expression, including *Jag2*, is suggested to be directly regulated by POGZ (Supplementary Fig. 13). Considering that POGZ is suggested to form part of a nuclear complex with CHD4 as well as SP1 and HP1[35,36,47], POGZ is likely to regulate transcriptional networks controlling neuronal differentiation through chromatin remodeling.

We determined that the ASD-associated de novo mutations of POGZ decreased the nuclear localization of the POGZ protein, impairing its function and compromising cortical excitatory neuron development (Figs. 1e–h and 4a, b). Post-mortem studies

have found that developmental abnormalities associated with neuronal migration can occur in ASD[2]; several ASD-associated gene products, such as CHD8, RELN, CNTNAP2, AUTS2, WDFY3, and TBR1, are differentially involved in excitatory neuron development[2,48]. Further studies will be important for elucidating the molecular link between altered excitatory neuron development and ASD phenotypes.

$POGZ^{WT/Q1038R}$ mice sufficiently recapitulated the pathogenic abnormalities in patients with NDDs (Fig. 5, Supplementary Figs. 8, 12). Developmentally, consistent with the observation that 13 out of 34 patients with NDDs who carry de novo POGZ mutations are diagnosed with microcephaly[16,28,30–34], $POGZ^{WT/Q1038R}$ mice showed smaller brain size than their WT littermates (Supplementary Fig. 8e–h). In the adult stage, in addition to behavioral abnormalities, excitatory neurons in the cerebral cortex were hyperactivated during the social interaction task in $POGZ^{WT/Q1038R}$ mice, leading to altered cellular E/I balance within neural circuitry. During development, decreased number of excitatory neurons may induce a compensatory increase in the activity of excitatory neurons in the adult brain. Given that POGZ is expressed at high levels in the embryonic brain both in humans (http://hbatlas.org/hbtd/images/wholeBrain/POGZ.pdf http://hbatlas.org/hbtd/images/nctxBrain/POGZ.pdf) and in mice (Fig. 2a–c), the disease-associated deficit in the adult stage may begin prenatally with impaired neuronal development not only in $POGZ^{WT/Q1038R}$ mice but also in patients with sporadic ASD who carry de novo mutations in POGZ[49]. The $POGZ^{WT/Q1038R}$ mouse is a good model for studying the links between prenatal deficits in cortical excitatory neuron development and clinically relevant abnormalities in adult brain function[50,51].

Consistent with the finding that the activity of excitatory neurons in the cerebral cortex was increased, the social deficits were rescued by administration of NBQX, a selective AMPA receptor antagonist in $POGZ^{WT/Q1038R}$ mice (Supplementary Fig. 16b). Furthermore, inspired by works showing that ASD and epilepsy have partial clinical and biological overlaps and share common molecular pathogenic mechanisms[52,53], we applied perampanel, an anti-epileptic negative allosteric modulator of the AMPA receptor and determined that the administration of perampanel also rescued the social deficits in $POGZ^{WT/Q1038R}$ mice (Supplementary Fig. 16d). Together with the fact an excitatory shift in the cellular E/I balance of the cerebral cortex is suggested to be associated with social and cognitive deficits[54,55], pharmaceutical modulation of glutamate signaling could be an effective therapeutic strategy for ASD patients with POGZ mutation.

In summary, by comprehensively examining these human and mouse materials, we biologically demonstrated POGZ-mediated regulation of transcriptional networks controlling neuronal differentiation and the damaging effects of de novo POGZ mutations identified in ASD patients. Importantly, despite the prenatal origin of impaired brain development in $POGZ^{WT/Q1038R}$ mice, impaired social interaction, a core symptom of ASD is pharmacologically treatable in adult $POGZ^{WT/Q1038R}$ mice. Together with iPSC lines obtained from the patient carrying the Q1042R POGZ mutation, $POGZ^{WT/Q1038R}$ mice are versatile tools for developing therapeutics for ASD as well as analyzing the molecular pathogenesis of ASD.

## Methods

**Ethics statement.** This study was carried out in accordance with the World Medical Association's Declaration of Helsinki and was approved by the Research Ethics Committee in Osaka University (#28-8-1). All recombinant DNA experiments were reviewed and approved by the Gene Modification Experiments Safety Committee at Osaka University (#04389). The animal experiments were performed in accordance with the guidelines for animal use issued by the Committee of Animal Experiments, Osaka University, Jikei University School of Medicine and RIKEN Tsukuba Branch, and were approved by the Committee in Osaka

**Table 1 In silico prediction of the effect of de novo POGZ missense mutations identified in patients with NDDs and unaffected controls.**

| Amino-acid change | PROVEAN | SIFT | PolyPhen2 | MutationTaster | CADD score | ACMG classification | Case |
|---|---|---|---|---|---|---|---|
| S314N | −0.70 (neutral) | 0.211 (tolerated) | 0.000 (benign) | 0.853 (polymorphism) | 17.25 | Likely pathogenic | ASD |
| Y597C | −7.01 (deleterious) | 0.000 (damaging) | 0.999 (probably damaging) | 0.997 (disease causing) | 26.60 | Likely pathogenic | ASD |
| H641Q | −7.66 (deleterious) | 0.001 (damaging) | 0.989 (probably damaging) | 1.000 (disease causing) | 25.30 | Likely pathogenic | ASD |
| S799N | −1.81 (neutral) | 0.014 (damaging) | 0.985 (probably damaging) | 0.999 (disease causing) | 26.10 | Likely Pathogenic | ASD |
| E1040K | −0.94 (neutral) | 0.003 (damaging) | 0.999 (probably damaging) | 1.000 (disease causing) | 31.00 | Pathogenic | ASD/ID |
| Q1042R | −0.92 (neutral) | 0.004 (damaging) | 0.991 (probably damaging) | 1.000 (disease causing) | 27.20 | Pathogenic | ASD |
| T1210P | −0.85 (neutral) | 0.038 (damaging) | 0.077 (benign) | 1.000 (polymorphism) | 13.48 | Likely pathogenic | Unclassified NDDs |
| D23G | −1.75 (neutral) | 0.001 (damaging) | 0.997 (probably damaging) | 1.000 (disease causing) | 26.90 | Uncertain significance | Control |
| N136S | −0.17 (neutral) | 0.009 (damaging) | 0.118 (benign) | 0.899 (disease causing) | 21.00 | Uncertain significance | Control |
| N159D | −1.39 (neutral) | 0.004 (damaging) | 0.993 (probably damaging) | 0.994 (disease causing) | 26.30 | Uncertain significance | Control |
| N341S | −0.47 (neutral) | 0.054 (tolerated) | 0.384 (benign) | 0.923 (polymorphism) | 16.72 | Uncertain significance | Control |
| R374Q | −0.70 (neutral) | 0.01 (damaging) | 0.993 (probably damaging) | 0.989 (disease causing) | 27.20 | Uncertain significance | Control |
| R379Q | −1.40 (neutral) | 0.031 (damaging) | 0.997 (probably damaging) | 0.955 (disease causing) | 26.20 | Uncertain significance | Control |
| P446L | −1.25 (neutral) | 0.204 (tolerated) | 1.000 (probably damaging) | 1.000 (disease causing) | 24.40 | Uncertain significance | Control |
| R502K | −0.17 (neutral) | 0.641 (tolerated) | 0.064 (benign) | 0.866 (disease causing) | 22.00 | Uncertain significance | Control |
| V59I | −6.15 (deleterious) | 0.003 (damaging) | 0.994 (probably damaging) | 0.999 (disease causing) | 24.20 | Uncertain significance | Control |
| N632S | 0.21 (neutral) | 0.137 (tolerated) | 0.882 (possibly damaging) | 0.897 (polymorphism) | 22.10 | Uncertain significance | Control |
| R674C | −2.53 (deleterious) | 0.03 (damaging) | 1.000 (probably damaging) | 1.000 (disease causing) | 32.00 | Uncertain significance | Control |
| R797Q | −1.70 (neutral) | 0.04 (damaging) | 0.997 (probably damaging) | 0.996 (disease causing) | 26.40 | Uncertain significance | Control |
| D828N | −3.68 (deleterious) | 0.247 (tolerated) | 0.835 (possibly damaging) | 1.000 (disease causing) | 23.80 | Uncertain significance | Control |
| R1005H | −0.75 (neutral) | 0.018 (damaging) | 0.998 (probably damaging) | 0.921 (disease causing) | 28.20 | Likely benign | Control |
| F1051L | −0.62 (Neutral) | 0.452 (tolerated) | 0.573 (possibly damaging) | 0.998 (disease causing) | 18.31 | Likely benign | Control |
| H1084R | −1.05 (neutral) | 0.319 (tolerated) | 0.666 (possibly damaging) | 0.999 (disease causing) | 24.90 | Likely benign | Control |
| L1103H | −0.46 (neutral) | 0.024 (damaging) | 0.875 (possibly damaging) | 0.573 (polymorphism) | 22.30 | Uncertain significance | Control |
| A1288V | −0.45 (neutral) | 0.013 (damaging) | 0.261 (benign) | 0.967 (disease causing) | 23.10 | Uncertain significance | Control |
| D1404N | −0.38 (neutral) | 0.001 (damaging) | 0.990 (probably damaging) | 0.582 (disease causing) | 27.60 | Uncertain significance | Control |

University, Jikei University School of Medicine and RIKEN Tsukuba Branch, respectively (Osaka University, #28-1-15; Jikei University School of Medicine, #2017-083; RIKEN Tsukuba Branch, #T2019-007).

**Antibodies.** The primary antibodies used for immunoblotting were rabbit anti-POGZ (SIGMA-Aldrich, #AV39172, 1:1000), mouse anti-Myc (9E10) (Santa Cruz Biotechnology, CA, USA, #sc-40, 1:400), rabbit anti-Lamin A/C (Cell Signaling Technology, MD, USA, #2032, 1:1000), and mouse anti-α-Tubulin (DM1A) (SIGMA-Aldrich, #T9026, 1:5000); the secondary antibodies used for immuno-blotting were horseradish peroxidase (HRP)-conjugated goat anti-rabbit IgG (Santa Cruz Biotechnology, #sc-2004, 1:1000), HRP-conjugated goat anti-mouse IgG (Santa Cruz Biotechnology, #sc-2005, 1:1000), alkaline phosphatase (AP)-conjugated goat anti-rabbit IgG (Santa Cruz Biotechnology, #sc-2007, 1:1000), and AP-conjugated goat anti-mouse IgG (Santa Cruz Biotechnology, #sc-2008, 1:1000). The primary antibodies used for immunostaining were rabbit anti-GFP (MBL, Aichi, Japan, #598, 1:200), chicken anti-GFP (Abcam, Cambridge, UK, #ab13970, 1:500), rabbit anti-PAX6 (BioLegend, CA, USA, #901301, 1:50), rat anti-SOX2 (Molecular Probe, OA, USA, #A-24339, 1:50), rabbit anti-TBR2 (Abcam, #ab23345, 1:50), mouse anti-SATB2 (Abcam, #ab51502, 1:50), rabbit anti-CUX1 (Santa Cruz Biotechnology, #sc-13024, 1:50), rat anti-CTIP2 (Abcam, #ab18465, 1:50), rabbit anti-TBR1 (Proteintech, IL, USA, #20932-1-AP, 1:50), rat anti-BrdU (Abcam, #ab6326, 1:40), rabbit anti-MAP2 (Merck Millipore, MA, USA, #AB5622, 1:200), and mouse anti-NESTIN (Merck Millipore, #MAB5326, 1:1000). The secondary antibodies used for immunostaining were biotinylated goat anti-rabbit IgG (Vector Labs, CA, USA, #BA-1000, 1:200), biotinylated goat anti-mouse IgG (Vector Labs, #BA-9200, 1:200), Alexa Fluor 488-conjugated goat anti-rabbit IgG (Life Technologies, CA, USA, #A-11008, 1:200), Alexa Fluor 488-conjugated goat anti-chicken IgY (Life Technologies, #A-11039, 1:500), Alexa Fluor 647-conjugated goat anti-rat IgG (Life Technologies, #A-21247, 1:200), and Alexa Fluor 594-conjugated donkey anti-mouse IgG (Jackson ImmunoResearch, PA, USA, #715-585-150, 1:250). The primary antibodies used for immunocytochemistry of ASD patient-derived and control iPSC lines were mouse anti-TRA-1-60 (1:1000), mouse anti-TRA-1-81 (1:500), rabbit anti-SOX2 (1:800), and rabbit anti-OCT-4A (1:800), which were included in the Stem Light Pluripotency Antibody Kit (Cell Signaling Technology, #9656S). The antibody used for fluorescence in situ hybridization (FISH) was peroxidase (POD)-conjugated sheep anti-Digoxigenin (DIG; Fab fragments; Roche Life Sciences, Basel, Switzerland, #11207733910, 1:250). The antibodies used for ChIP were rabbit anti-POGZ (Bethyl Laboratories, TX, USA, A302-509A), rabbit anti-POGZ (Bethyl Laboratories, A302-510A) and normal rabbit IgG (Merck Millipore, 12-370).

**Neuro2a cell culture and transfection.** Mouse neuroblastoma Neuro2a cells (ATCC CCL-131) were cultured in Dulbecco's modified Eagle's medium (DMEM) supplemented with high glucose, GlutaMAX (Life Technologies), and 10% fetal bovine serum. Neuro2a cells were transfected using GenJet In Vitro Transfection Reagent for Neuro-2A Cells (Ver. II) (SignaGen Laboratories, MD, USA). The cells were fixed with 4% PFA in PBS for 10 min at room temperature or harvested and lysed with radio-immunoprecipitation assay buffer or a Cytoplasmic & Nuclear Protein Extraction Kit (101Bio, CA, USA) 3 days after transfection.

**SHSY-5Y cell culture and transfection.** Human neuroblastoma SHSY-5Y cells (ATCC CRL-2266) were cultured in DMEM with low glucose (Nissui, Tokyo, Japan) supplemented with 4 mM L-glutamine and 10% fetal bovine serum. SHSY-5Y cells were transfected using Lipofectamine 3000 Reagent (Thermo Fisher Scientific, MA, USA). The cells were harvested and lysed with a Cytoplasmic & Nuclear Protein Extraction Kit (101Bio) 2 days after transfection.

**Immunocytochemistry.** Immunocytochemistry were performed as previously described[56]. Briefly, cells were fixed with 4% PFA in PBS for 10 min at room temperature. Cells were permeabilized with PBS containing 0.1% Triton X-100 (Wako, Osaka, Japan) and incubated with blocking solution containing 4% normal goat serum (Thermo Fisher Scientific) in PBS for 1 h at room temperature, and then incubated with the blocking solution combined with primary antibodies overnight at 4 °C. The following day, the cells were incubated with the blocking solution combined with fluorescent-dye-conjugated secondary antibody and Hoechst 33258 dye (Calbiochem, CA, USA) for 1 h at room temperature[56]. For the immunocytochemistry of Neuro2a cells transfected with Myc-tagged mouse POGZ, Alexa Fluor 546-phalloidin (Molecular Probe) were used for staining of F-Actin. Images of the stained cells were acquired using an Olympus FluoView FV1000 confocal microscope (Olympus, Tokyo, Japan) and a BZ-9000 microscope (Keyence, Osaka, Japan). The images were then analyzed with ImageJ software (NIH, MD, USA) and Adobe Photoshop CS (Adobe Systems, CA, USA).

**Immunoblotting.** Lysates were resolved on 6–7.5% polyacrylamide–SDS gel by SDS–PAGE and transferred to polyvinylidene difluoride membranes[56]. Subsequently, these membranes were probed with the indicated primary antibodies overnight at 4 °C, followed by incubation with the indicated secondary antibodies for 1 h at room temperature. Proteins were visualized by AP-reaction using CDP star (Roche Life Sciences) and HRP-reaction using Western Lightning Plus ECL

(PerkinElmer, MA, USA). Data acquisition and analysis were performed using an LAS4000 image analyzer (GE Healthcare, NJ, USA). We have provided the uncropped blots in the Source Data file.

**Assay for nuclear localization of mutant POGZ.** Cytosolic and nuclear fractions from Neuro2a cells expressing Myc-tagged WT or mutant POGZ were prepared using a Cytoplasmic & Nuclear Protein Extraction Kit (101Bio) according to the manufacturer's protocol. Those fractions were subjected to immunoblotting with antibodies against Myc, Lamin A/C (a nuclear marker), and α-Tubulin (a cytosolic marker). The nuclear localization of POGZ was calculated as the ratio of the band intensity of Myc-POGZ in the nuclear fraction to that of total Myc-POGZ in the cytosolic and nuclear fractions combined.

**Reverse transcription and real-time PCR.** Total RNAs from cultured cells and tissues were isolated using the PureLink RNA Micro Kit (Thermo Fisher Scientific) and PureLink RNA Mini Kit (Thermo Fisher Scientific) according to the manufacturer's instructions. The total RNAs were reverse transcribed with Superscript III (Life Technologies). Real-time PCR was performed with SYBR Premix Ex Taq (Takara Bio Inc., Shiga, Japan) using CFX96 real-time PCR detection system (Bio-Rad Laboratories, CA, USA) as described previously[15]. The expression levels of *Pogz* (forward primer sequence: 5′-CCCTACCTATGTGCATTGTTCTC-3′; reverse primer sequence: 5′-TCCGTGGAACATGATTGTTG-3′) were normalized to those of *Gapdh* and were determined according to the $2^{-\Delta\Delta Ct}$ method.

**Immunohistochemistry.** E16.5, E17.5, and E18.5 mouse brains were fixed with 4% paraformaldehyde (PFA) in PBS overnight at 4 °C. The brains were sectioned at a 20 μm thickness by using a cryostat (Leica, Wetzlar, Germany, CM1520). Brains from 10-week-old adult mice were perfused with 4% PFA in PBS and post-fixed with 4% PFA in PBS overnight at 4 °C. The brains were sectioned at a 20 μm thickness by using a cryostat (Leica) for CUX1 staining after antigen retrieval methods or sectioned at a 50 μm thickness with a LinearSlicer PRO7N (DOSAKA EM CO.,LTD., Kyoto, Japan) for GABA staining. The brain slices were permeabilized with blocking solution containing 0.25% Triton X-100 (Wako), 1% normal goat serum (Thermo Fisher Scientific), and 1% bovine serum albumin (SIGMA-Aldrich) in PBS for 1 h at room temperature, and then incubated with the blocking solution combined with primary antibodies. The following day, the slices were incubated with the blocking solution combined with biotin-dye or fluorescent-dye-conjugated secondary antibody and Hoechst 33258 dye (Calbiochem) for 1 h at room temperature. The biotinylated secondary antibody was labeled with Texas Red-conjugated streptavidin (Vector Labs, #SA-5006). Three coronal sections per brain were imaged for quantification. Images of the stained brain slices were acquired using an Olympus FluoView FV1000 confocal microscope (Olympus) and a BZ-9000 microscope (Keyence). The images were then analyzed with ImageJ software (NIH) and Adobe Photoshop CS (Adobe Systems). The distributions of SATB2+ cells in E18.5 embryo brains and of CUX1+ cells and GABA+ cells in 10-week-old mice brains were quantified by dividing the cerebral wall into 10 equal bins (cortical plate (CP) 1 to intermediate zone (IZ)/SVZ 10).

**Fluorescence in situ hybridization.** For fresh samples, E16.5 mouse embryos were rapidly frozen using dry ice. The samples were cut at a thickness of 20 μm by using a cryostat (Leica) and collected on Matsunami adhesive silane (MAS)-coated glass slides (Matsunami Glass Ind., Ltd, Osaka, Japan). A cRNA probe sequence targeting *Pogz* (NCBI Reference Sequence: NM_172683.3) from base 1062 to base 1563 was amplified from mPOGZ cDNA (DNAFORM; clone ID: 30745658) via PCR and subcloned into a pBluescript II KS (+/−) vector. A 5′-digoxigenin (DIG)-labeled probe for *Pogz* was prepared by transcribing the *Bam*HI-linearized plasmid using T3 RNA polymerase (Roche Life Sciences). FISH for *Pogz* was performed as follows. The sections were fixed with 4% PFA in PBS for 30 min at room temperature. The slices were incubated with hybridization buffer containing the probe, and hybridization was allowed to take place overnight at 48 °C. On day 2, the slices were washed and incubated with 1% blocking buffer (Roche Life Sciences) containing anti-DIG-POD antibody at 4 °C overnight. On day 3, the anti-DIG-POD antibody was labeled with fluorescein using TSA Plus Fluorescein System (PerkinElmer). Immunostaining for each cell marker, including PAX6, SOX2, and TBR2, was performed as described above after FISH for *Pogz* at day 3. Images of the stained brain slices were acquired using an Olympus FluoView FV1000 confocal microscope (Olympus) and a BZ-9000 microscope (Keyence). The images were then analyzed with ImageJ software (NIH) and Adobe Photoshop CS (Adobe Systems).

**In utero electroporation.** In utero electroporation was performed on E14.5 embryos from timed-pregnant WT ICR mice (SLC, Shizuoka, Japan)[57]. The pregnant mice were anesthetized by intraperitoneal injections with a solution containing 0.3 mg medetomidine (Domitor, Zenoaq Nippon Zenyaku Kogyo, Fukushima, Japan), 4 mg midazolam (Dormicum, Astellas Pharma Inc., Tokyo, Japan), and 5 mg butorphanol (Bettlefar, MP AGRO Co., Ltd., Hokkaido, Japan) per kg. The uterine horns were exposed and the plasmid (2.5 μg/μL) mixed with Fast Green (0.1 mg/mL, SIGMA-Aldrich) were injected into the lateral ventricles. For the knockdown of *Pogz* and overexpression of WT-POGZ or POGZ mutants,

MISSION shRNA constructs or miR30-based shRNA constructs (1 μg/μL) and pcDNA3 expression constructs encoding WT-POGZ or POGZ mutants (1 μg/μL) were injected into the lateral ventricles together with a pCAG-GFP vector (0.5 μg/μL) expressing GFP at a 2:2:1 ratio. For the overexpression of WT-POGZ or POGZ mutants, pcDNA3 expression constructs encoding WT-POGZ or POGZ mutants (2 μg/μL) were injected into the lateral ventricles together with a pCAG-GFP vector (0.5 μg/μL) expressing GFP at a 4:1 ratio. Subsequently, electric pulses (35 V, 4 cycles; 50 ms on, 950 ms off) were applied to the head of the embryos targeting the dorsal-medial cortex using Square Wave Electroporator (NEPA GENE, Chiba, Japan, CUY21SC). Each pool of constructs was injected into three or four different embryos in at least two independent operations. The embryos were harvested 48 or 96 h later. Three non-adjacent coronal sections per brain were imaged for quantification. The images were acquired with an Olympus FluoView FV1000 (Olympus) confocal microscope and analyzed with ImageJ software (NIH) and Adobe Photoshop CS (Adobe Systems). In order to analyze the migration of GFP+ cells, embryos were dissected at E16.5, E17.5, or E18.5, and brain samples from the embryos were immunostained for GFP. Hoechst 33258 dye (Calbiochem) was used to stain the nuclei. The distribution of the GFP+ cells was quantified by dividing the cerebral wall into the CP, IZ, and SVZ/VZ at E16.5 or the upper CP, lower CP, and IZ/SVZ at E18.5. In order to assess the neuronal differentiation of GFP+ cells, the percentages of GFP+ PAX6+, GFP+TBR2+, and GFP+ SATB2+ cells was quantified.

**MISSION shRNA, miR30-based shRNA and POGZ expression constructs.** The MISSION shRNA TRC1 vectors (Supplementary Table 2) used in this study were purchased from Sigma-Aldrich (MO, USA). The sequence of shPOGZ^miR30 (Supplementary Table 3) was designed using an online design tool (http://cancan.cshl.edu/RNAi_central/RNAi.cgi?type = shRNA). The template oligonucleotide carrying the shPOGZ^miR30 sequence was obtained from Eurofins Genomics (Tokyo, Japan), amplified via PCR and subcloned into a pCAG-miR30 vector. In order to generate a plasmid expressing mPOGZ, mPOGZ cDNA (DNAFORM, Kanagawa, Japan, clone ID: 30745658) was amplified via PCR and subcloned into a pcDNA-6Myc vector. Plasmids expressing mPOGZ mutants were generated using a KOD mutagenesis kit (Toyobo, Osaka, Japan) according to the manufacturer's protocol.

**Subjects.** A patient with ASD was recruited from outpatient services at Osaka University Hospital. She had been diagnosed by at least two trained child psychiatrists according to the criteria in the Diagnostic and Statistical Manual of Mental Disorders, fourth edition, text revision (DSM-IV-TR)[58] based on interviews with the patient and unstructured or semi-structured observations of her behavior. During the interview, the pervasive developmental disorders Autism Society Japan Rating Scale (PARS)[59] and the Japanese version of the Asperger's Questionnaire (AQ)[60] were used to assist in the evaluation of ASD-specific behaviors and symptoms. The PARS is a semi-structured interview that is composed of 57 questions in eight domains corresponding to the characteristics of children with pervasive developmental disorders (PDDs), which was developed by the Autism Society Japan. The clinicians who diagnosed the subject were trained in the use of the PARS. Intelligence quotient data were collected using the full-scale Wechsler Adult Intelligence Scale (WAIS-III)[61]. Written informed consent was obtained from the subject after the procedures were fully explained.

P1381: The patient was a 16-year-old Japanese female. She met the criteria for PDD-Not Otherwise Specified (NOS), which refers to ASD. The patient had no physical disease. There was no specific abnormality in the results of her brain MRI or blood test. The patient's PARS and AQ scores were both above the cut-off point. Her scores on the WAIS-III were as follows: Full-scale IQ, 65; Verbal IQ, 70; Performance IQ, 65; Verbal Comprehension Index, 73; Working Memory Index, 90; Perceptual Organization Index, 65; and Processing Speed Index, 78. A delay in the patient's communication skills had been noted at a check-up when she was 18 months old. She could not communicate well with others of similar age and screeched if she was forced to go to school. She had grammatical language impairment, specifically in subject–predicate relations, and had sound sensitivity as well. She persisted in asking about specific subjects or activities that she interested her.

P1399: The subject was a 49-year-old Japanese male, the father of the patient.

**Generation of iPSC lines.** iPSC lines were generated using immortalized B cells obtained from the ASD patient carrying Q1042R POGZ mutation and from her unaffected father[62]. Plasmid vectors for induction of pluripotency, including 0.63 μg pCE-hOCT3-4, 0.63 μg pCE-hSK, 0.63 μg pCE-hUL, 0.63 μg pCE-mp53DD, and 0.50 μg pCXB-EBNA1 (Addgene, MA, USA) were electroporated into the immortalized B cells using the Nucleofector 2b Device (Lonza, Basel, Switzerland) with the Amaxa Human T-cell Nucleofector Kit (Lonza). The immortalized B cells introduced with the reprogramming factors were cultivated in Roswell Park memorial Institute (RPMI) medium (Wako) containing 10% fetal bovine serum for 1 week. Subsequently, the electroporated immortalized B cells were seeded onto mitomycin C-treated mouse SL10 feeder cells (ReproCELL Inc., Kanagawa, Japan) and cultured for 20–30 days in DMEM/F12 (Thermo Fisher Scientific) containing 20% KnockOut Serum Replacement (Thermo Fisher

Scientific), 1% non-essential amino acid solution (SIGMA-Aldrich), 2 mM L-glutamine (Thermo Fisher Scientific), 0.1 mM 2-mercaptoethanol (Wako), and 4 ng/mL bFGF (PeproTech, NJ, USA). Colonies of cells similar to human embryonic stem (ES) cells were clonally isolated, morphologically selected, subjected to PCR-based analysis of episomal vector loss and evaluated for expression of pluripotent markers using immunocytochemistry (Oct-4A, Sox2, TRA-1-60, and TRA-1-81).

**Neural induction of iPSC lines and expansion of NSCs.** The neural induction of iPSC lines into NSCs was performed with PSC Neural Induction Medium (Thermo Fisher Scientific) according to the manufacturer's instructions. For neural induction, iPSC lines were cultured in Essential 8 medium (Thermo Fisher Scientific) under feeder-free conditions on Matrigel (Corning, NY, USA). On day 0 of neural induction, ~24 h after the cells were split, the culture medium was replaced with PSC neural induction medium containing neurobasal medium and PSC neural induction supplement. The neural induction medium was exchanged every other day from day 0 to day 4 of neural induction and every day after day 4 of neural induction. On day 7 of neural induction, NSCs (P0) were harvested and expanded in neural expansion medium containing 50% neurobasal medium (Thermo Fisher Scientific), 50% advanced DMEM/F12 (Thermo Fisher Scientific), and neural induction supplement (Thermo Fisher Scientific) on Matrigel. Expanded NSCs after passage 6 were used for subsequent assays.

**Neuronal differentiation assay of NSCs derived from iPSC lines.** ASD-patient-derived and control NSCs were seeded in Brain Phys basal medium (Nacalai Tesque, Kyoto, Japan)-based neuronal differentiation medium containing 1% N2 supplement (Wako), 2% B27 supplement (Thermo Fischer Scientific), 200 μM L-ascorbic acid (Sigma-Aldrich), 1 mM dBcAMP (Wako), 20 ng/mL human brain-derived neurotrophic factor (BDNF; R&D Systems, MN, USA), 20 ng/mL human glial-cell-line-derived neurotrophic factor (GDNF; R&D Systems), 500 ng/mL mouse laminin (Thermo Fisher Scientific), and 1 μM (3,5-difluorophenylacetyl)-L-alanyl-L-2-phenylglycine tert-butyl ester (Wako) on a 24-well plate coated with poly-L-ornithine (Sigma-Aldrich), 6.67 μg/mL human fibronectin (Thermo Fisher Scientific), and 6.67 μg/mL mouse laminin (Thermo Fisher Scientific). On day 2, which was in an early stage of neuronal differentiation, the cells were fixed with 4% PFA for 15 min at room temperature. Immunocytochemistry was performed as described previously[56]. Briefly, the cells were permeabilized with 0.1% Triton X-100 (Wako) in PBS for 10 min and incubated with blocking solution containing 1% normal goat serum in PBS for 1 h at room temperature. Then, the cells were incubated with the blocking solution combined with primary anti-MAP2 antibody overnight at 4 °C. The following day, the cells were incubated with secondary Alexa Fluor 488 conjugated antibody for 1 h at room temperature. The images were acquired using a ToxInsight automated microscope (Thermo Fisher Scientific). The proportion of MAP2+ neurons was automatically analyzed using the same microscope.

**BrdU ELISA.** ASD-patient-derived and control NSCs were seeded in neural expansion medium. The following day, the culture medium was replaced with neural expansion medium containing 10 μM BrdU. BrdU+ proliferating cells were labeled with POD-conjugated anti-BrdU antibody using a Cell Proliferation ELISA BrdU Kit (Roche Life Sciences) 3 h after the addition of the BrdU. The absorbance was measured using an iMARK Microplate Reader (Bio-Rad Laboratories) 15 min after treatment with a substrate of POD (3,3′5,5′-Tetramethylbenzidine).

**Proliferation and migration assays of neurospheres.** ASD-patient-derived and control NSC suspension in DMEM/F-12 (Sigma-Aldrich) containing 15 mM HEPES buffer (Sigma-Aldrich), 2% B27 supplement (Thermo Fisher Scientific), 20 ng/mL human epidermal growth factor (EGF; PeproTech), 20 ng/mL human basic fibroblast growth factor (bFGF; PeproTech), 10 ng/mL human leukemia inhibitory factor (LIF; Merck Millipore), and 0.1% heparin sodium solution 1000 IU/5 mL (Nippon Zenyaku Kogyo, Fukushima, Japan) were placed in a 96-well V-bottom plate. The NSCs proliferated to form neurospheres in suspension culture. In order to analyze the proliferation of NSCs, the diameter of neurospheres cultured for 5 days was measured. For the in vitro migration assay, neurospheres cultured for 3 days were transplanted into neuronal differentiation medium on a six-well plate coated with poly-L-ornithine (Sigma-Aldrich), 6.67 μg/mL human fibronectin (Thermo Fisher Scientific), and 6.67 μg/mL mouse laminin (Thermo Fisher Scientific). On day 2 of neuronal differentiation, cell migration was evaluated by measuring the distance from the edge of the neurosphere to the nucleus of the most distant cell.

**Generation of *POGZ*<sup>WT/Q1038R</sup> mice.** The *POGZ*<sup>WT/Q1038R</sup> mouse strain was generated using a genome-editing technique as described previously[63] with some modifications. The sequence 5′-GCAGCAGCTCCCTGTAAATG-3′ was selected as the target for single guide RNA (sgRNA). Cas9 mRNA and sgRNA were produced using linearized T7-NLS hCas9-pA (RIKEN BioResource Research Center (BRC) #RDB13130, a gift from Tomoji Mashimo[64] and DR274 (Addgene #42250, a gift from Keith Joung)[65], respectively. A 117-nt single-stranded oligodeoxynucleotide (ssODN) donor coding the POGZ Q1038R mutation was ordered as Ultramer DNA oligos from Integrated DNA Technologies (Skokie, IL, USA). The

Cas9 mRNA, sgRNA, and ssODN were dissolved in nuclease-free water and microinjected into the cytoplasm of pronuclear-stage mouse zygotes obtained from C57BL/6NJcl mice (CLEA Japan, Tokyo, Japan). The genotype was determined by PCR analysis using genomic DNA derived from the mice with the mouse WT *Pogz* (forward primer sequence: 5′-TCTGTGAAGAAGCTTCGGGTAGTAC-3′; reverse primer sequence: 5′-GTCTCCTCATTTACAGGGAGCT-3′); and mouse Q1038R *Pogz* (forward primer sequence: 5′-GCAGCGGCTCCCCGTTAAC-3′; reverse primer sequence: 5′-AGCGCACAGCCCACTCATAG-3′). POGZWT/Q1038R mice will be available from BRC as strain number RBRC09544.

**Hematoxylin–eosin (H&E) staining.** Brains from 10-week-old adult male mice were perfused with 4% PFA in PBS and post-fixed with 4% PFA in PBS overnight at 4 °C. The brains were sectioned at a 50 μm thickness. The brain slices at Bregma 0 mm were stained by HE staining using Mayer's Hemalum Solution (Merck Millipore) and Eosin Y-solution 0.5% aqueous (Merck Millipore) according to the manufacturer's instructions. Peripheral organs, including eye, cochlea, trachea, stomach, duodenum, ileum, cecum, and colon, from 10-week-old adult male mice were perfused with 10% formalin neutral buffer solution (Wako). HE staining for the slices of the peripheral organs from WT and *POGZ*<sup>WT/Q1038R</sup> mice was performed by New Histo. Science Laboratory Co., Ltd.

**Micro-CT analysis.** Mouse skulls and embryos were scanned using a micro-CT scanner (ScanXmate-E090S, Comscantecno Co., Ltd., Kanagawa, Japan). For skull analysis, the head region of 22-week-old male mice were scanned at a tube voltage of 80 kV and a tube current of 50 μA. We used TRI/3D-BON (RATOC Systems, Osaka, Japan) and Image J software (NIH) for reconstructing the three-dimensional images of skulls and measuring the linear distances between landmarks, respectively. Before embryo scanning, E15.5 mouse embryos were fixed with Bouin solution (Wako) at room temperature, stored in 70% ethanol and then treated with 1% phosphotungstic acid. These samples were scanned at a tube voltage of 40 kV and a tube current of 100 μA. Section images were analyzed using the OsiriX (www.osirix-viewer.com) software.

**BrdU labeling in *POGZ*<sup>WT/Q1038R</sup> mice.** New born neurons were labeled with BrdU at E14.5 by intraperitoneal administration of BrdU to pregnant female mice at a dosage of 50 mg/kg. In order to analyze the migration of BrdU+ cells, embryos were dissected at E18.5 and brain samples from the embryos were immunostained for BrdU and SATB2. Three non-adjacent coronal sections per brain were imaged for quantification. The images were acquired with an Olympus FluoView FV1000 (Olympus) confocal microscope and analyzed with ImageJ software (NIH) and Adobe Photoshop CS (Adobe Systems). The distribution of the BrdU+ SATB2+ cells was quantified by dividing the cerebral wall into 10 equal bins (CP 1–IZ/SVZ 10).

**Mouse cortical NSCs cultures.** The suspension of cortical cells prepared from E16.5 embryonic mouse cortex in NSCs expansion medium containing DMEM/F-12 (Thermo Fisher Scientific), 1% N2 supplement (Wako), 20 ng/mL EGF (PeproTech), and 20 ng/mL bFGF (PeproTech) was placed into a six-well plate. The NSCs were expanded to form primary neurospheres in suspension culture for a week. The neurospheres were dissociated into single cells using trypsin (Thermo Fisher Scientific) and seeded in the NSCs expansion medium on Matrigel. The secondary NSCs were expanded in adhesion. When the cells reached 80% confluency 2 days after plating, the cells were harvested and used in the following experiments.

**RNA sequence.** RNA sequence and the following analysis of the aligned reads was performed as described previously[62]. Total RNAs isolated from NSCs differentiated from the iPSC lines and mouse cortical secondary NSCs were sequenced using the Illumina HiSeq2000 system (BGI, Beijing, China) and the Illumina HiSeq2500 system (Genome Information Research Center, Osaka University, Osaka, Japan), respectively. The expression levels of each gene were analyzed based on Fragments Per Kilobase of exon per Million mapped (FPKM).

**GO and pathway analysis.** The ToppGene Suite (https://toppgene.cchmc.org/) was used for GO annotation-based functional classification of differentially expressed genes. The genes with |fold change| ≥ 1.2 were included for the analysis of Biological Process (gene in annotation: 1000 ≤ n ≤ 100,000). The GO annotations were cut off at $P < 0.05$ and FDR < 0.05.

**ChIP assay.** Chromatin was isolated from secondary NSCs derived from E16.5 cortex of WT mice. ChIP was performed using ChIP-IT Express Enzymatic Magnetic ChIP Kit & Enzymatic Shearing Kit (Active Motif, CA, USA) according to the manufacturer's instructions. Briefly, NSCs at 80% confluency on 15 cm dish were crosslinked in DMEM/F12 containing 1% formaldehyde. Following nuclear extraction, chromatin was sheared by enzyme for 30 min at 37 °C. The sheared chromatin was immunoprecipitated with the antibody in a solution containing protein-G magnetic beads overnight at 4 °C. The immunoprecipitated chromatin was eluted from the beads and the crosslinking was reversed. Following ChIP, the

DNA samples were purified using Chromatin IP DNA Purification Kit (Active Motif) and amplified via PCR using GenoMatrix Whole Genome Amplification Kit (Active Motif). Quantified PCR (qPCR) was performed using the amplified DNA with the primers for the mouse *Jag2* promoter region (forward primer 1 sequence: 5′-GGATGGCCTGATTGTGTGT-3′; reverse primer 1 sequence: 5′-TTCGGAG-GAGGGGTCTTC-3′; forward primer 2 sequence: 5′-AGATGGGAAGACCCCT CCT-3′; reverse primer 2 sequence: 5′-CAGTGCCACAGAGGGTTACA-3′; forward primer 3 sequence: 5′-GTCATGGGGATCCAGCTTT-3′; reverse primer 3 sequence: 5′-TCCCAGGCCTTTATACCACA-3′)[66,67]. The levels of enrichment of each amplicon by ChIP were normalized to the amount of amplified DNA fragments immunoprecipitated with each antibody.

**Behavioral tests.** All behavioral tests were carried out on male C57BL/6NJcl mice at 1.5–4 months of age, except the ultrasonic vocalization test and the juvenile play test. All behavioral experiments were performed during the light period by experimenters who were blind to the genotypes and treatments of the mice.

Home cage activity: Each mouse was placed in a test cage under a 12 h:12 h light–dark cycle (light on at 8:00). After 1 day of habituation, spontaneous activity in the light and dark phases, total activity and the ratio of activity in the light phase to total activity were measured for 5 days using an infrared activity sensor (O'Hara & Co., Ltd., Tokyo, Japan)[68].

Fear conditioning test: The fear-conditioning test was performed using an Image FZ4 (O'Hara & Co., Ltd.), automated fear contextual and tone-dependent fear conditioning apparatus[68]. On the training day (day 1), each mouse was placed into a shock chamber (Box A; $10 \times 10 \times 10$ cm, white polyvinyl chloride boards, stainless steel rod floor, O'Hara & Co., Ltd.) for 120 s, immediately followed by the presentation of four tone–shock pairs at 90 s intervals. Each tone–shock pair included a tone (70 dB, 10 kHz) for 30 sc and a 0.5 mA foot shock lasting for the final 0.5 s of the tone. On day 2, in order to quantify contextual freezing, each mouse was placed back in box A for 6 min of measurement. On day 3, each mouse was placed in a white transparent chamber (Box B) for 120 s, and then the 30 s tone was presented four times at 90 s intervals. Freezing during the first 120 s was measured as pre-tone freezing, and freezing during the tone presentations was measured as cued freezing.

Novel object recognition test: Prior to each session, mice were acclimatized to the test room for at least 30 min. After 10 min of habituation to the experimental box under dim lighting conditions (10–20 lx) for 3 consecutive days, the test mouse was allowed to freely explore two novel objects (A and B) in the box for 10 min. Twenty-four hours after the training session, the retention session was conducted. In the retention session, object B was replaced with novel object C, and the mouse was allowed to move freely for 10 min in the same box. The exploration time for each object was measured. The discrimination index (%) was the difference between the exploration time for the novel object and that for the familiar object divided by total exploration time. This index was used to calculate values for recognition memory. This test was conducted between 10:00 and 14:00.

Open field test: Locomotor activity was measured using the open field. Each mouse was placed in the center of the open-field apparatus ($45 \times 45 \times 30$ cm). The total distance traveled and time spent in the center area ($25 \times 25$ cm) were recorded. Data were collected for 90 min per mouse.

Light/dark transition test: The light/dark transition test was performed using an apparatus that consisted of two sections of equal size ($20 \times 20 \times 25$ cm, O'Hara & Co., Ltd.)[68]. The illumination was 353 lx in the light chamber and 0.1 lx in the dark chamber. Each mouse was placed in the middle of the light chamber and allowed to move freely. The total distance traveled, latency to enter the light chamber, time spent in the light chamber and number of transitions between chambers were automatically measured using Image LD4 (O'Hara & Co., Ltd.) for 10 min.

Y-maze test: Each mouse was placed on the center of an apparatus consisting of three arms (arm length: 40 cm, arm bottom width: 3 cm, arm upper width: 10 cm, height of wall: 12 cm, O'Hara & Co., Ltd.)[68]. The distance traveled and the alteration ratio were measured.

PPI test: Each mouse was habituated to a sound-proof box ($33 \times 43 \times 33$ cm, O'Hara & Co., Ltd.) with 65 dB background noise for 5 min. In order to acclimatize mice to the startle pulse, 110 dB per 40 ms of white noise was presented during the 5 min habituation period, and the startle response to this white-noise stimulus was excluded from the statistical analysis. A prepulse sound of 70, 75, 80, and 85 dB was presented for 20 ms, followed by the presentation of 110 dB for 50 ms. These prepulse and startle sounds were presented 10 times in pseudorandom order. Startle amplitude was measured 50 ms after presentation of the prepulse sound. The percentage of PPI was calculated as [(startle amplitude without prepulse)−(startle amplitude of trial with prepulse)]/(startle amplitude without prepulse) × 100[68].

Reciprocal social interaction test: A male intruder mouse was placed in the test cage after habituation of the test mouse to the same cage for 60 min. Over the full experimental period (20 min), the total duration of time that the resident mouse spent sniffing the intruder mouse was measured. The drug administration was performed 30 min after the beginning of habituation. This test was carried out between 10:00 and 14:00.

Self-grooming test: Prior to the self-grooming test, mice were acclimatized to the testing room for 60 min. This test was conducted in a new mouse cage without bedding. The cumulative time that the test mouse spent grooming itself was measured for 10 min following 10 min period of habituation to the test cage.

Juvenile playing test: The juvenile playing test was carried out on juvenile male mice at 3 weeks of age, before weaning. Each test mouse was placed alone in the test cage for a 60 min habituation period. A male non-sibling intruder mouse from a different litter but similar in body weight to the test mouse (±1.0 g) was placed in the test cage after habituation. Over the full experimental period (10 min), the total durations of various behavioral events including sniffing, allogrooming, chasing, and playing were measured. This test was carried out between 10:00 and 14:00.

USV test: On postnatal day 4, the pups and their mothers were acclimatized to the test room for at least 30 min. Each male pup was removed from the home cage containing the mother and littermates and placed in a clean glass 500 mL Pyrex beaker with paper-towel bedding. The USVs of each pup were detected for 3 min using an UltraSoundGateCM16/CMPA microphone (Avisoft Bioacoustics, Glienicke, Germany) in a sound attenuation chamber under light control (90 lx), and recorded with Avisoft-SASLab Pro software (Avisoft Bioacoustics). The microphone was placed in the box ~20 cm above the pup, and USVs between 40 and 150 kHz were measured. The waveform patterns of the calls were analyzed by classifying the calls into 10 distinct categories, based on internal pitch changes, lengths, and shapes according to a previous report[69].

**FAST whole-brain imaging.** Ten-week-old adult male WT- and *POGZ*[WT/Q1038R]-Arc-dVenus mice were perfused with 4% PFA in PBS 5 h after the reciprocal social interaction test, at the peak of the dVenus expression driven by Arc promoter. The mouse brain was embedded in 4% agarose gel (Nacalai Tesque) dissolved in PBS. Subsequently, serial whole-brain imaging was performed using FAST[44,45]. The FAST whole-brain images were obtained at a resolution of $1.0 \times 1.0 \times 5$ μm$^3$ with an sCMOS camera (Andor Technology, Belfast, UK) with a $2 \times 2$ binning mode, a ×16 NA 0.8 objective lens (Nikon instruments, Tokyo, Japan), and a ×0.83 intermediate magnification lens (Yokogawa Electronic, Tokyo, Japan). Semiautomatic anatomical parcellation of the brain and quantification of dVenus-positive neurons in each brain region were performed using TRI/FCS-NUC64 software (Tatoc System Engineering, Tokyo, Japan).

**Principal component analysis (PCA).** PCA was performed in R (https://www.r-project.org/) using the preinstalled prcomp function. The data were normalized before PCA by dividing the number of dVenus-positive cells in each brain region by the total number of dVenus-positive cells in all brain regions.

**Golgi impregnation and dendritic spine analysis.** Golgi impregnation was performed on 10-week-old WT and *POGZ*[WT/Q1038R] mice with the FD Rapid GolgiStain Kit (FD Neuro Technologies, MD, USA), according to the manufacturer's protocol. Fully focused images were obtained using a BZ-9000 microscope (Keyence) and quantitatively analyzed using ImageJ software (NIH).

**Electrophysiology.** WT and *POGZ*[WT/Q1038R] mice (10–11 week-old, male) were decapitated under isoflurane anesthesia (5% in 100% O$_2$) and the brains were quickly removed and secured on the cutting stage of a vibrating blade slicer (VT1200S, Leica). Coronal slices (300 μm thick) containing the ACC were cut in an ice-cold cutting solution composed of (in mM) 2.5 KCl, 0.5 CaCl$_2$, 10 MgSO$_4$, 1.25 NaH$_2$PO$_4$, 2 thiourea, 3 sodium pyruvate, 92 N-methyl-D-glucamine, 20 HEPES, 12 N-acetyl-L-cysteine, 25 D-glucose, 5 L-ascorbic acid and 30 NaHCO$_3$ equilibrated with 95% O$_2$ + 5% CO$_2$ (pH ~ 7.4; osmolality, ~280 mOsm/kg) and incubated in the cutting solution at 34 °C for 15–20 min. The slices were then kept at room temperature (20–25 °C) in the standard artificial cerebrospinal fluid (ACSF) composed of (in mM) 125 NaCl, 3 KCl, 2 CaCl$_2$, 1.3 MgCl$_2$, 1.25 NaH$_2$PO$_4$, 10 D-glucose, 0.4 L-ascorbic acid, and 25 NaHCO$_3$ (pH 7.4 bubbled with 95% O2 + 5% CO$_2$; osmolality, ~310 mOsm/kg) until the electrophysiological recording. Each slice was transferred to a recording chamber (~0.4 mL volume) and fixed with nylon grids attached to a platinum frame. The slice was submerged and continuously superfused at a rate of 1.5–2.5 mL/min with the standard ACSF at 30–32 °C. Whole cell membrane current was recorded from the pyramidal neurons in the layer II/II of the ACC visually identified under an upright microscope (BX-51WI, Olympus) with oblique illumination. Patch-clamp electrodes (4–6 MΩ) were made from borosilicate glass pipettes (1B150F-4, World Precision Instruments) and filled with internal solution containing (in mM) 122.5 potassium gluconate, 10 HEPES, 17.5 KCl, 0.2 EGTA, 8 NaCl, 2 MgATP, 0.3 NaGTP (pH, 7.2; osmolarity, 290–300 mOsm). The membrane potential was held at −60 mV. The membrane current was recorded with a MultiClamp 700B amplifier (Molecular Devices), filtered at 2 kHz and digitized at 10 kHz with a 16-bit resolution using a PowerLab interface (AD Instruments). mEPSCs were recorded in the presence of picrotoxin (100 μM) and tetrodotoxin (1 μM) and analyzed by Igor Pro 7 (WaveMetrics). 100 events in each neuron were quantified. All experiments were performed in a manner blinded to the mouse genotype during the experiments and analyses.

**Drug administration.** NBQX disodium salt hydrate (Abcam) dissolved in saline was intraperitoneally administered to 10-week-old male mice at a dosage of 10 mg/kg 30 min before the open field test or reciprocal social interaction test. Perampanel

powder (Toronto Research Chemicals, ON, Canada) was suspended in a 0.5% weight-per-volume methyl cellulose (400 cP, Wako) solution. Perampanel was administered via oral gavage to 10-week-old male mice at a dose of 3 mg/kg in a volume of 20 μL/g 30 min before the open field test or reciprocal social interaction test.

**Statistical analysis.** The quantified data from the western blots and qRT-PCR were statistically analyzed using one-way ANOVA followed by Bonferroni–Dunn post hoc tests. The quantified data from the immunohistochemistry and HE staining analysis were statistically analyzed using two-way ANOVA with repeated measures followed by Bonferroni–Dunn post hoc tests and Student's t-test. The quantified data from in utero electroporation were statistically analyzed using one-way ANOVA and two-way ANOVA with repeated measures followed by Bonferroni–Dunn post hoc tests. The quantified data from the neuronal differentiation assay, the BrdU ELISA on NSCs and the proliferation and migration assays on neurospheres were statistically analyzed using Student's t-test. The body weights of WT and POGZ^WT/Q1038R mice were statistically analyzed using two-way ANOVA with repeated measures followed by Bonferroni–Dunn post hoc tests. The micro-CT data were statistically analyzed using Welch's t-test. The behavioral data were statistically analyzed using one-way ANOVA and two-way ANOVA followed by Bonferroni–Dunn post hoc tests. The Golgi staining data were statistically analyzed using Student's t-test. The electrophysiological data were statistically analyzed using Mann–Whitney U test and Kolmogorov–Smirnov test. For details, see the description in each figure legend. The significance level was set at $P < 0.05$. Statistical analyses were conducted using Stat-View (SAS Institute, NC, USA) and R (version 3.4.1).

**In silico prediction of the effect of missense mutations.** The pathogenicity of NDD-related and control de novo missense mutations and their effect on the POGZ function were predicted using PROVEAN, SIFT (http://provean.jcvi.org/protein_batch_submit.php?species = human), PolyPhen2 (http://genetics.bwh.harvard.edu/pph2/), MutationTaster (http://www.mutationtaster.org/), CADD score (https://cadd.gs.washington.edu/snv), and The American College of Medical Genetics and Genomics (ACMG) classification[70].

**Reporting summary.** Further information on research design is available in the Nature Research Reporting Summary linked to this article.

## Data availability
RNA sequencing data have been deposited to the DDBJ Sequence Read Archive (DRA) and are available at the accession number DRA009486. The source data underlying Fig. 1e, g and i and Supplementary Figs. 3a, 4b, 4f, 7b, and 8d are provided as a Source Data file. All data supporting the finding of this study are available with the Article and its Supplementary Information or from the corresponding author upon the reasonable request.

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

## Acknowledgements

We thank Dr. Masataka Kikuchi for helpful discussion. We acknowledge the NGS core facility of the Genome Information Research Center at the Research Institute for Microbial Diseases of Osaka University for the support in RNA sequencing and data analysis. This work was supported in part by JSPS KAKENHI, grant numbers JP15H04645 (T.N.), JP18H02574 (T.N.), JP17K19488 (H.H.), JP17H03989 (H.H.), JP16K07004 (A.M.W.) and JP17H05960 (A.M.W.); the JSPS Research Fellowships for Young Scientists, grant number JP17J00152 (K.M.); JST CREST, grant number JPMJCR1751 (A.M.W.); MEXT KAKENHI, grant numbers JP18H05416 (H.H.), JP19H05217 (A.K.), JP19H04909 (T.N.) and JP19H05218 (T.N.); AMED, grant numbers JP19gm1310003 (T.N.), JP18dm0107122 (H.H.), JP18dm0207061 (H.H.), and JP18am0101084; Nagai Memorial Research Scholarship, Pharmaceutical Society of Japan (K.M.); and grants from the Takeda Science Foundation (T.N.), the Japan Foundation for Pediatric Research (T.N.), the Asahi Glass Foundation (T.N.), and the Pharmacological Research Foundation, Tokyo (T.N.). This study was also supported in part by Center for Medical Research and Education, Graduate School of Medicine, Osaka University.

## Author contributions

K.M., R.H., H.H., and T.N. designed the experiments and wrote the manuscript. K.M., S.O., K.N., K.Y., K.K., H.M., N.G-N., M.B., M.K., K.U., A.H-T., N.S., T.I., M.S., H.O., K.T., H.H. and T.N. designed and performed the biochemical and cellular experiments and analyzed the data. M.N. and A.M.W. performed the electrophysiological experiments and analyzed the data. S.A., I.Y., T.F., M.T., S.W. and A.Y. generated POGZ^{WT/Q1038R} mice, performed the behavioral analyses and analyzed the data. H.S. and M.T. performed micro-CT analysis. K.M., S.H., Y.A. and T.N. designed and performed the behavioral analyses and analyzed the data. K.S., H.I., M.H., A.K., S.Y., and H.H. designed and performed the imaging and analyzed the data. Y.Y., H.Y., M.F. and R.H. recruited and characterized the patient with ASD and performed the experiments with clinical samples. All authors read and approved the manuscript.

## Competing interests

H.O. is a founding scientist of SanBio Co. Ltd. and K Pharma Inc. The other authors declare no competing interests.

## Additional information

Kensuke Matsumura[1,2,3], Kaoru Seiriki[1,2], Shota Okada[1], Masashi Nagase[4], Shinya Ayabe [5], Ikuko Yamada[6], Tamio Furuse[6], Hirotoshi Shibuya[6], Yuka Yasuda[7,8], Hidenaga Yamamori[7,9], Michiko Fujimoto[7,10], Kazuki Nagayasu[1], Kana Yamamoto[1], Kohei Kitagawa[1], Hiroki Miura[1], Nanaka Gotoda-Nishimura[1], Hisato Igarashi[1], Misuzu Hayashida[1], Masayuki Baba[1], Momoka Kondo[1], Shigeru Hasebe[11], Kosei Ueshima[11], Atsushi Kasai [1], Yukio Ago [1,12], Atsuko Hayata-Takano [1,13], Norihito Shintani [1], Tokuichi Iguchi[14], Makoto Sato [14,15,16], Shun Yamaguchi[17,18], Masaru Tamura [6], Shigeharu Wakana[6,19], Atsushi Yoshiki[5], Ayako M. Watabe [4], Hideyuki Okano[20], Kazuhiro Takuma [11,13], Ryota Hashimoto[7,21], Hitoshi Hashimoto [1,13,22,23,24✉] & Takanobu Nakazawa [1,11✉]

[1]Laboratory of Molecular Neuropharmacology, Graduate School of Pharmaceutical Sciences, Osaka University, Suita, Osaka 565-0871, Japan. [2]Interdisciplinary Program for Biomedical Sciences, Institute for Transdisciplinary Graduate Degree Programs, Osaka University, Suita, Osaka 565-0871, Japan. [3]Research Fellowships for Young Scientists of the Japan Society for the Promotion of Science, Chiyoda-ku, Tokyo 102-0083, Japan. [4]Institute of Clinical Medicine and Research, Jikei University School of Medicine, Kashiwa, Chiba 277-8567, Japan. [5]Experimental Animal Division, RIKEN BioResource Research Center, Tsukuba, Ibaraki 305-0074, Japan. [6]Technology and Developmental Team for Mouse Phenotype Analysis, RIKEN BioResource Research Center, Tsukuba, Ibaraki 305-0074, Japan. [7]Department of Pathology of Mental Diseases, National Institute of Mental Health, National Center of Neurology and Psychiatry, Kodaira, Tokyo 187-8553, Japan. [8]Life Grow Brilliant Clinic, Osaka, Osaka 530-0012, Japan. [9]Japan Community Health care Organization Osaka Hospital, Osaka, Osaka 553-0003, Japan. [10]Department of Psychiatry, Graduate School of Medicine, Osaka University, Suita, Osaka 565-0871, Japan. [11]Department of Pharmacology, Graduate School of Dentistry, Osaka University, Suita, Osaka 565-0871, Japan. [12]Laboratory of Biopharmaceutics, Graduate School of Pharmaceutical Sciences, Osaka University, Suita, Osaka 565-0871, Japan. [13]Molecular Research Center for Children's Mental Development, United Graduate School of Child Development, Osaka University, Kanazawa University, Hamamatsu University School of Medicine, Chiba University and University of Fukui, Suita, Osaka 565-0871, Japan. [14]Department of Anatomy and Neuroscience, Graduate School of Medicine, Osaka University, Suita, Osaka 565-0871, Japan. [15]United Graduate School of Child Development, Osaka University, Kanazawa University, Hamamatsu University School of Medicine, Chiba University and University of Fukui, Suita, Osaka 565-0871, Japan. [16]Research Center for Child Mental Development, University of Fukui, Fukui, Fukui 910-1193, Japan. [17]Department of Morphological Neuroscience, Gifu University Graduate School of Medicine, Gifu 501-1194, Japan. [18]Center for Highly Advanced Integration of Nano and Life Sciences, Gifu University, Gifu 501-1194, Japan. [19]Department of Gerontology, Institute of Biomedical Research and Innovation, Kobe, Hyogo 650-0047, Japan. [20]Department of Physiology, Keio University School of Medicine, Shinjuku-ku, Tokyo 160-8582, Japan. [21]Osaka University, Suita, Osaka 565-0871, Japan. [22]Division of Bioscience, Institute for Datability Science, Osaka University, Suita, Osaka 565-0871, Japan. [23]Transdimensional Life Imaging Division, Institute for Open and Transdisciplinary Research Initiatives, Osaka University, Suita, Osaka 565-0871, Japan. [24]Department of Molecular Pharmaceutical Science, Graduate School of Medicine, Osaka University, Suita, Osaka 565-0871, Japan. ✉email: hasimoto@phs.osaka-u.ac.jp; takanobunakazawa-tky@umin.ac.jp

