## [Peer Review File · Nature Communications]

Reviewers' Comments:

Reviewer #2:

Remarks to the Author:

The study of Matsumara et al, provides a comprehensive investigation of the role of the ASD/ID related gene POGZ in neurodevelopment and adult brain function. The study includes several highly technical experimental approaches and the findings support a novel and critical role for POGZ mutations in neuronal development. The paper is generally well written and would be of interest to a broad scientific community.

I have specific comments below which mainly pertain to the disconnect between experiments presented in Figures 1-3 (study of the effects of specific mutant POGZ sequences) and those related to the mouse and human iPSC work (study of the Q1042R mutation only).

1. Results: Page 7. The authors state 'After a series of biochemical experiments' they determined that mutant POGZ shows less nuclear localization compared to non-disease related mutations and WT. The authors should state explicitly what experimental techniques were used in this section to derive these data.
2. 'POGZ regulates the development of mouse cortical neural stem cells' in this section of the results the authors show that the migration of GFP+ cells in the developing brain was inhibited by POGZ knockdown. These findings are clearly interesting and provide critical information with regards to the effects of POGZ deficiency, but it is unclear how they relate biologically to the observations of 'reduced nuclear localization' of POGZ related to disease-associated mutations. Although this is somewhat addressed with the mutant overexpression rescue experiments in Figure 4, it would appear that single experiments showing the effect expression of mutant POGZ would be more appropriate.
3. How do the data derived from iPSC cell line work of an individual with the Q1042R POGZ mutation relate to experiments in Figures 1-3?. It does not appear that this mutation was investigated prior to this point in any of the preceding experiments. Inclusion of this mutant sequence in the rescue experiments in Fig 4 seems essential, especially given that the mouse transgenic is also modelling the Q1042R mutation.
4. Page 13 results: 'In the other tests, including the light/dark transition, Y-maze and PPI tests, POGZWT/Q1038R mice showed behavioral features similar to those of their WT littermates (data not shown)'. The authors should present the PPI results and Y-maze data.
5. Fig 7. How were the doses of NBQX and Per determined? Did either drug affect activity in the open field? Since alterations in locomotor activity could impact the sociability test, it is imperative that these experiments are conducted.
6. Supplementary Table 2 is over 600 pages long. This is not acceptable and should be edited to show only significant gene results.

Reviewer #3:

Remarks to the Author:

This manuscript endeavors to identify how different variants in the POGZ gene may result in various neurodevelopmental phenotypes. The authors have employed a multi-modal approach that is novel and very informative. The manuscript is clearly written and well-organized. The manuscript could be improved by a more rigorous classification/description of genetic variants, as well as clarifying other minor points.

1. The authors discuss variants in POGZ found in a "normal" cohort as well as those with neurodevelopmental phenotypes. It would be helpful to describe these variants in a more consistent way, such as employing the classification system promulgated by the American College of Medical

Genetics and Genomics (see PMID# 25741868).

2. In the manuscript and in Figure 1, the authors parse the phenotypes of ASD, ID and White-Sutton syndrome, however the method for assignment into these not described. Please clearly describe how phenotyping was done (including what neurodevelopmental assessments for ASD and IQ were done and dysmorphology or other evaluations) and what criteria were utilized to assign subjects to these different groups.

3. It is stated that no mice homozygous for the p.Q1038R were identified and presumably this is lethal. Were embryos studied to provide more information about this (i.e. were there specific embryonic developmental abnormalities that lead to lethality)? If not, could this be done? As a corollary, were heterozygous mice evaluated for other non-neurological abnormalities seen in humans (e.g. diaphragmatic hernia, craniofacial abnormalities, gastrointestinal abnormalities, etc.)?

4. I am unclear from the manuscript if the impaired neuronal migration affected all parts of the brain equally. It would be helpful to specify what areas were assessed (and not assessed) and whether there were differences. Also, it appears that the thickness of cortical layers is not different from wildtype, but I did not find specific measurements to demonstrate this. It would be helpful to provide measurements of the thickness of various layers in heterozygotes and wild type animals. Also, was the gyral pattern normal and were there other findings (heterotopias, etc.) of abnormal neuronal migration?

5. In Table 1, rather than providing multiple different in silico measures of variant pathogenicity, it would be more helpful to provide a CADD score; also, as mentioned previously, employing the ACMG variant classification rubric would be helpful in understanding these variants.

6. There are many figures; I do not find figures 5 and 7 essential to understanding the manuscript and would suggest moving these to the supplemental figures section.

7. In the legend for Figure 3b, the abbreviations for the cortical layers are not spelled out as they are in Figures 2 & 4 - please spell out.

8. The manuscript is longer than the standard for the journal (an article is asked to be capped at 6500 words and 50 references).

Responses to reviewers

We thank the reviewers for critically reading our manuscript and raising several important issues. We addressed all the comments by the reviewers and followed their suggestions as far as possible. We feel that the manuscript has been significantly improved. We have indicated the changes in the text with red so that they are easily identifiable.

Responses to Reviewer #2:

The study of Matsumura et al provides a comprehensive investigation of the role of the ASD/ID related gene POGZ in neurodevelopment and adult brain function. The study includes several highly technical experimental approaches and the findings support a novel and critical role for POGZ mutations in neuronal development. The paper is generally well written and would be of interest to a broad scientific community. I have specific comments below which mainly pertain to the disconnect between experiments presented in Figures 1-3 (study of the effects of specific mutant POGZ sequences) and those related to the mouse and human iPSC work (study of the Q1042R mutation only).

We appreciate Reviewer #2's positive evaluation of our work and thank her/him for constructive suggestions. Our point-by point responses to the comments are as follows.

[Reviewer's comment #1]

1) Results: Page 7. The authors state 'After a series of biochemical experiments' they determined that mutant POGZ shows less nuclear localization compared to non-disease related mutations and WT. The authors should state explicitly what experimental techniques were used in this section to derive these data.

[Our response to comment #1]

We apologize for the ambiguous descriptions. Since previous studies have suggested that POGZ is localized to the nucleus and functions as a chromatin regulator, we assume that the ASD-related *de novo* mutations may alter the nuclear localization of POGZ. To examine this possibility, prior to the fractionation experiments (Fig. 1 in the previous and revised manuscript), we conducted immuno-cytochemical experiments using ASD-related missense mutants, E1036K- and Q1038R-mutated POGZ, as well as E1043X-mutated POGZ, the longest nonsense-mutated POGZ, and found that these mutations partially impaired the nuclear localization of POGZ (Supplementary Fig. 2 in the revised manuscript (new figure in the revised manuscript)). We have amended the relevant description as follows.

Page 7, lines 19-25

Previous studies have suggested that POGZ is localized to the nucleus and functions as a chromatin regulator, we therefore assume that the ASD-related *de novo* mutations may alter the nuclear localization of POGZ. To examine this possibility, we firstly conducted immuno-cytochemical experiments using ASD-related missense mutants, E1036K (E1040K in human POGZ)- and Q1038R (Q1042R in human POGZ)-mutated POGZ, as well as E1043X (E1047X in human POGZ)-mutated POGZ, the longest nonsense-mutated POGZ, and found that these mutations partially impaired the nuclear localization of POGZ (Supplementary Fig. 2).

[Reviewer's comment #2]

2) 'POGZ regulates the development of mouse cortical neural stem cells' in this section of the results the authors show that the migration of GFP+ cells in the developing brain was inhibited by POGZ knockdown. These findings are clearly interesting and provide critical information

with regards to the effects of POGZ deficiency, but it is unclear how they relate biologically to the observations of ‘reduced nuclear localization’ of POGZ related to disease-associated mutations. Although this is somewhat addressed with the mutant overexpression rescue experiments in Figure 4, it would appear that single experiments showing the effect expression of mutant POGZ would be more appropriate.

[Our response to comment #2]

We thank the reviewer for raising a very important issue. As suggested by the reviewer, we performed overexpression experiments using ASD-related *de novo* mutated POGZ using wild-type embryos. We determined that the expression of R1004X-, E1036K-, Q1038R- and E1043X-mutated POGZ impaired the migration of GFP⁺ cells in wild-type neurons (Supplementary Fig. 5 in the revised manuscript (new figure in the revised manuscript)), suggesting that the *de novo* mutations show a dominant-negative effect upon cell migration. Considering that the *de novo* mutated POGZ showed reduced nuclear localization (Fig.1), abnormally localized *de novo* mutated POGZ in the cytoplasm might inhibit the function of endogenous POGZ (e.g., abnormally localized *de novo* mutated POGZ may titrate the interaction partner of POGZ in the cytoplasm). Alternatively or in addition, wild-type and *de novo* mutated POGZ might compete each other in the nucleus. We have added the relevant description as follows.

Page 10, lines 7-15

We then performed overexpression experiments using ASD-related *de novo* mutated POGZ using wild-type embryos. We determined that the expression of R1004X-, E1036K-, Q1038R- and E1043X-mutated POGZ impaired the migration of GFP⁺ cells in wild-type neurons (Supplementary Fig. 5), suggesting that the *de novo* mutations show a dominant-negative effect upon cell migration. Considering that the *de novo* mutated POGZ showed reduced nuclear localization (Fig. 1), abnormally localized *de novo* mutated POGZ in the cytoplasm might inhibit the function of endogenous POGZ (e.g., abnormally localized *de novo* mutated POGZ may titrate the interaction partner of POGZ in the cytoplasm). Alternatively or in addition, wild-type and *de novo* mutated POGZ might compete each other in the nucleus.

[Reviewer’s comment #3]

3) How do the data derived from iPSC cell line work of an individual with the Q1042R POGZ mutation relate to experiments in Figures 1-3?. It does not appear that this mutation was investigated prior to this point in any of the preceding experiments. Inclusion of this mutant sequence in the rescue experiments in Fig 4 seems essential, especially given that the mouse transgenic is also modelling the Q1042R mutation.

[Our response to comment #3]

We thank the reviewer for raising a very important issue. We used the mouse POGZ cDNA and its mutants, including Q1038R mouse-POGZ, for the experiments shown in Fig.1-3 in the original manuscript, in which we used mouse Neuro2a cells and mouse embryos. Since the amino acid sequences of the human and mouse POGZ are very similar (93.9% identify in amino acid sequence) (Supplementary Fig. 1 in the revised manuscript (new figure in the revised manuscript)), we think that the mouse Q1038R mutation is likely to corresponding to the human Q1042R mutation and that the data from the experiments with the mouse Q1038R mutation in Fig. 1-3 are related to the data using iPSC from the patient with human Q1042R POGZ mutation. We sincerely hope that the reviewer will agree with us on this point. To further connect between the data in Fig. 1-3 and human iPSC work, we performed cellular fractionation

experiments using human SH-SY5Y cells and the human Q1042R-mutated POGZ and obtained essentially the same results as the results with the mouse Q1038R mutation in Fig. 1 (Supplementary Fig. 3 in the revised manuscript (new figure in the revised manuscript)). We have added the relevant descriptions as follows.

Page 7, lines 16-18

Since the amino acid sequences of the human and mouse proteins are very similar (93.9 % identify in amino acid sequence) (Supplementary Fig. 1), we think that each mouse mutation is likely to corresponding to the respective human mutation.

Page 8, lines 12-14

Additionally, we performed cellular fractionation experiments using human SH-SY5Y cells and the human Q1042R-mutated POGZ and obtained essentially the same results as the results with the mouse Q1038R mutation in Fig. 1g, h (Supplementary Fig. 3).

[Reviewer's comment #4]

4)Page 13 results: 'In the other tests, including the light/dark transition, Y-maze and PPI tests, POGZ^{WT/Q1038R} mice showed behavioral features similar to those of their WT littermates (data not shown)'. The authors should present the PPI results and Y-maze data.

[Our response to comment #4]

We followed the reviewer's suggestion and presented the PPI results and Y-maze data (Supplementary Fig. 14h-k in the revised manuscript (new figures in the revised manuscript)).

[Reviewer's comment #5]

5) Fig 7. How were the doses of NBQX and Per determined? Did either drug affect activity in the open field? Since alterations in locomotor activity could impact the sociability test, it is imperative that these experiments are conducted.

[Our response to comment #5]

We thank the reviewer for raising an important issue. We hypothesized that the elevated activation of excitatory neurons may be involved in impaired social interaction in POGZ^{WT/Q1038R} mice and that suppression of the elevated activation may rescue the impaired social interaction. To examine the possibility, according to the previous studies (Araki et al., Int. J. Neuropsychopharmacol., 2014; Filliat et al., Pharmacol. Biochem. Behav., 1998; Harada et al., Epilepsia, 2011; Kato et al., Nat. Med., 2016), we determined the minimum doses of NBQX and perampanel for antiepileptic activity. With these doses, we found that 10 mg/kg of NBQX did not affect the locomotor activity and that 3 mg/kg of perampanel tended to slightly decrease the locomotor activity in the open field, which is not statistically significant (Supplementary Fig. 16a and c in the revised manuscript (new figures in the revised manuscript)). While the locomotor activity tended to slightly decrease, perampanel as well as NBQX effectively rescued the impaired social interaction in POGZ^{WT/Q1038R} mice (Supplementary Fig. 16b and d in the revised manuscript (Fig. 7l and m in the previous manuscript)). We have added the relevant descriptions as follows.

Page 16, lines 15-19

According to the previous studies, we determined the minimum doses of NBQX and perampanel for antiepileptic activity. With these doses, we found that 10 mg/kg of NBQX did

not affect the locomotor activity and that 3 mg/kg of perampanel tended to slightly decrease the locomotor activity in the open field, which is not statistically significant (Supplementary Fig. 16a, c).

[Reviewer's comment #6]

6) Supplementary Table 2 is over 600 pages long. This is not acceptable and should be edited to show only significant gene results.

[Our response to comment #6]

We followed the reviewer's suggestion and edited Supplementary Table 3 and 4 in the previous manuscript to show commonly differentially expressed genes annotated to "neurogenesis (GO: 0022008)" between human and mouse (*i.e.*, 78 out of 913 and 251 genes annotated to GO: 0022008 in human and mouse, respectively showed commonly differential expression between human and mouse) (Supplementary Table 4 and 5 in the revised manuscript (new tables in the revised manuscript)). We have amended the relevant descriptions as follows.

Page 13, line 11

(Supplementary Table 3, 4) was changed to (Significant results are shown in Supplementary Table 4, 5)

We have added the relevant descriptions as follows.

Page 13, lines 15-18

In particular, 78 out of 913 and 251 genes annotated to neurogenesis (GO: 0022008) in human and mouse, respectively showed commonly differential expression between human and mouse (Supplementary Table 4, 5).

We wish to thank reviewer #2 again for his/her pertinent critique and very helpful suggestions.

Responses to Reviewer #3:

This manuscript endeavors to identify how different variants in the POGZ gene may result in various neurodevelopmental phenotypes. The authors have employed a multi-modal approach that is novel and very informative. The manuscript is clearly written and well-organized. The manuscript could be improved by a more rigorous classification/description of genetic variants, as well as clarifying other minor points.

We appreciate Reviewer #3's critical evaluation on our work and thank her/him for pointing out important issues. Our point-by-point responses to the comments are as follows.

[Reviewer's comment #1]

1) The authors discuss variants in POGZ found in a "normal" cohort as well as those with neurodevelopmental phenotypes. It would be helpful to describe these variants in a more consistent way, such as employing the classification system promulgated by the American College of Medical Genetics and Genomics (see PMID# 25741868).

[Our response to comment #1]

We thank the reviewer for raising a very important issue. We followed the reviewer's suggestion and edited Table 1 to show ACMG classification, in which our current results regarding E1040K, Q1042R, R1005H, F1051L and H1084R were reflected (Table 1 in the previous and revised manuscript).

[Reviewer's comment #2]

2) In the manuscript and in Figure 1, the authors parse the phenotypes of ASD, ID and White-Sutton syndrome, however the method for assignment into these not described. Please clearly describe how phenotyping was done (including what neurodevelopmental assessments for ASD and IQ were done and dysmorphology or other evaluations) and what criteria were utilized to assign subjects to these different groups.

[Our response to comment #2]

We thank the reviewer for raising an important issue. In this study, we classified patients into ASD, ID and White-Sutton syndrome according to the original diagnosis in each report (Neale et al, Nature, 2012; Iossifov et al, Nature, 2014; Rubeis et al, Nature, 2014; DDD study, Nature, 2015; Fukai et al, J Hum Genet, 2015; Sanders et al, Neuron, 2015; Ye et al, Cold Spring Harb Mol Case Stud, 2015; Hashimoto et al, J Hum Genet, 2016; Stessman et al, Am J Hum Genet, 2016; Tan et al, J Hum Genet, 2016; White et al, Genome Med, 2016; DDD study, Nature, 2017; Dentici et al, Am J Med Genet A, 2017; Yuen et al, Nat Neurosci, 2017). We found that diagnostic information is not described in a part of these reports. Accordingly, instead of diagnostic information of each patient, we provided the detailed information of patients, including subject ID and reference for instant access to patients' phenotypes (Supplementary Table 1 in the revised manuscript (new table in the revised manuscript)). The relevant description is in page 6, lines 1-2.

Page 5, line 25-page 6, line 2

(see Fig. 1a and Supplementary Table 1; we classified patients into ASD, ID and White-Sutton syndrome according to the original diagnosis in each report.)

[Reviewer's comment #3]

3) It is stated that no mice homozygous for the p.Q1038R were identified and presumably this is lethal. Were embryos studied to provide more information about this (i.e. were there specific embryonic developmental abnormalities that lead to lethality)? If not, could this be done? As a corollary, were heterozygous mice evaluated for other non-neurological abnormalities seen in humans (e.g. diaphragmatic hernia, craniofacial abnormalities, gastrointestinal abnormalities, etc.)?

[Our response to comment #3]

We thank the reviewer for raising an important issue. To answer this comment, we performed additional experiments as follows.

- 1) We performed micro-CT scanning of mouse embryos and found that homozygous $POGZ^{Q1038R/Q1038R}$ mouse embryos showed a ventricular septal defect, which likely results in embryonic lethality (Supplementary Fig. 11b in the revised manuscript (new figure in the revised manuscript)). Importantly, all embryos examined in this study showed a ventricular septal defect (n = 4).
- 2) We histologically examined patient-related non-neurological abnormalities in adult heterozygous $POGZ^{WT/Q1038R}$ mice and found that $POGZ^{WT/Q1038R}$ mice did not exhibit any significant changes in peripheral organs, including eye, cochlea, trachea, stomach, duodenum, ileum, caecum and colon, compared to WT mice (Supplementary Fig. 9 in the revised manuscript (new figure in the revised manuscript)). Additionally, we did not find diaphragmatic hernia in adult heterozygous $POGZ^{WT/Q1038R}$ mice (data not shown). Furthermore, we performed micro-CT scanning of adult heterozygous $POGZ^{WT/Q1038R}$ mice. We did not find any significant abnormalities in the skull of $POGZ^{WT/Q1038R}$ mice (Supplementary Fig. 10 in the revised manuscript (new figure in the revised manuscript)).

We have added the relevant descriptions as follows.

Page 12, lines 5-8

We performed micro-CT scanning of mouse embryos and found that $POGZ^{Q1038R/Q1038R}$ mouse embryos (E15.5) showed a ventricular septal defect, which likely results in embryonic lethality (n = 4) (Supplementary Fig. 11b).

Page 11, line 23-page 12, line 3

We histologically examined patient-related non-neurological abnormalities in adult $POGZ^{WT/Q1038R}$ mice and found that $POGZ^{WT/Q1038R}$ mice did not exhibit any significant changes in peripheral organs, including eye, cochlea, trachea, stomach, duodenum, ileum, caecum and colon, compared to WT mice (Supplementary Fig. 9). Additionally, we did not find diaphragmatic hernia in adult $POGZ^{WT/Q1038R}$ mice (data not shown). Furthermore, we performed micro-CT scanning of adult $POGZ^{WT/Q1038R}$ mice. We did not find any significant abnormalities in the skull of $POGZ^{WT/Q1038R}$ mice (Supplementary Fig. 10).

[Reviewer's comment #4]

4) I am unclear from the manuscript if the impaired neuronal migration affected all parts of the brain equally. It would be helpful to specify what areas were assessed (and not assessed) and whether there were differences. Also, it appears that the thickness of cortical layers is not different from wildtype, but I did not find specific measurements to demonstrate this. It would be helpful to provide measurements of the thickness of various layers in heterozygotes and wild

type animals. Also, was the gyral pattern normal and were there other findings (heterotopias, etc.) of abnormal neuronal migration?

[Our response to comment #4]

We thank the reviewer for raising an important issue. In this study, we examined neuronal migration in the developing somatosensory cortex and found that POGZ knockdown impaired neuronal migration in the developing somatosensory cortex (Fig. 2, 3 in the revised manuscript). We then measured the thickness of cortical layers in the somatosensory cortex and found that the thicknesses of layer II-IV and V in POGZ^{WT/Q1038R} mice were slightly decreased and increased, respectively (Supplementary Fig. 8i-n in the revised manuscript (new figures in the revised manuscript)), which might be attributable to the abnormal distribution of layer II/III neurons (Fig. 2). Furthermore, although POGZ^{WT/Q1038R} mice exhibited decreased brain size, we did not find any drastic histological abnormalities, such as heterotopias, in the cortex of POGZ^{WT/Q1038R} mice. We have amended the relevant descriptions as follows.

Page 9, line 5

The electroporated embryos were allowed to develop until E18.5 and histologically analyzed for migration of GFP+ cells in the developing cortex.

was changed to

The electroporated embryos were allowed to develop until E18.5 and histologically analyzed for migration of GFP+ cells in the developing somatosensory cortex.

Page 9, lines 18-19

We analyzed the proportion of GFP+ NSCs, IPs and neurons at E16.5 (2 days after *in utero* electroporation) and determined that *Pogz* knockdown increased the proportion of PAX6+ NSCs and decreased the proportion of TBR2+ IPs and SATB2+ young neurons without significantly affecting migration within 2 days (Fig. 3a-h).

was changed to

We analyzed the proportion of GFP+ NSCs, IPs and neurons at E16.5 (2 days after *in utero* electroporation) and determined that *Pogz* knockdown increased the proportion of PAX6+ NSCs and decreased the proportion of TBR2+ IPs and SATB2+ young neurons without significantly affecting migration in the somatosensory cortex within 2 days (Fig. 3a-h).

We have added the relevant descriptions as follows.

Page 11, lines 18-23

We next measured the thickness of cortical layers in the somatosensory cortex and found that the thicknesses of layer II-IV and V in POGZ^{WT/Q1038R} mice were slightly decreased and increased, respectively (Supplementary Fig. 8i-n). Although POGZ^{WT/Q1038R} mice exhibited decreased brain size, we did not find any drastic histological abnormalities, such as heterotopias, in the cortex of POGZ^{WT/Q1038R} mice.

[Reviewer's comment #5]

5) In Table 1, rather than providing multiple different *in silico* measures of variant pathogenicity, it would be more helpful to provide a CADD score; also, as mentioned

previously, employing the ACMG variant classification rubric would be helpful in understanding these variants.

[Our response to comment #5]

We thank the reviewer for raising an important issue. We followed the reviewer's suggestion and provided the CADD score as well as ACMG variant classification (Table 1 in the revised manuscript).

[Reviewer's comment #6]

6) There are many figures; I do not find figures 5 and 7 essential to understanding the manuscript and would suggest moving these to the supplemental figures section.

[Our response to comment #6]

We followed the reviewer's suggestion and moved Fig. 5 and 7 in the previous manuscript to Supplementary Fig. 8, 12, 15 and 16 in the revised manuscript.

[Reviewer's comment #7]

7) In the legend for Figure 3b, the abbreviations for the cortical layers are not spelled out as they are in Figures 2 & 4 - please spell out.

[Our response to comment #7]

We apologize for the omissions. We spelled out the abbreviations for the cortical layers in the legend for Fig. 3a, b as follows in the revised manuscript.

Legends for Fig 3a, b

a, Slight, non-significant migration defects caused by shRNA-mediated knockdown of *Pogz* in E16.5 mouse cortices electroporated at E14.5. Scale bars, 50 μ m. **b**, Quantification of GFP⁺ cells in each layer (each n = 4).

was changed to

a, Slight, non-significant migration defects caused by shRNA-mediated knockdown of *Pogz* in E16.5 mouse cortices electroporated at E14.5 (CP, cortical plate; IZ, intermediate zone; SVZ, subventricular zone; VZ, ventricular zone). Scale bars, 50 μ m. **b**, Quantification of GFP⁺ cells in each layer (CP, cortical plate; IZ, intermediate zone; SVZ, subventricular zone; VZ, ventricular zone) (each n = 4).

[Reviewer's comment #8]

8) The manuscript is longer than the standard for the journal (an article is asked to be capped at 6500 words and 50 references).

[Our response to comment #8]

We thank the reviewer for raising a constructive comment. According to the Guide to Authors of *Nature Communications*, the main text (not including Abstract, Methods, References and Figure legends) is limited to 5,000 words, and references should not exceed 70. The number of words of the main text is 4,606, which meets the regulation. Regarding references, we deleted

non-essential references (ref. 69, 70, 71, 72, 74 and 75 in the previous manuscript) for following the regulation (70 references in the revised manuscript). In addition, for shortening our manuscript, we moved Fig. 5 and 7 in the previous manuscript to **Supplementary Fig. 8, 12, 15 and 16** (please see our response to the comment #6). Taken together, we think that the length of our manuscript could be acceptable. We sincerely hope that the reviewer will agree with us on this point.

We wish to thank reviewer #3 again for his/her pertinent critique and very helpful suggestions.

Reviewers' Comments:

Reviewer #2:

Remarks to the Author:

In the revision, the authors have competently and successfully addressed my issues and concerns and included additional experiments to support their findings. I have no further comments.

Reviewer #3:

None